# *HiMaCon:* Discovering Hierarchical Manipulation Concepts from Unlabeled Multi-Modal Data

**Ruizhe Liu**[1]    **Pei Zhou**[1]    **Qian Luo**[1,4]    **Li Sun**[1]
**Jun Cen**[3]    **Yibing Song**[3]    **Yanchao Yang**[1,2]

[1]HKU Musketeers Foundation Institute of Data Science, The University of Hong Kong
[2]Department of Electrical and Electronic Engineering, The University of Hong Kong
[3]DAMO Academy, Alibaba Group    [4]Transcengram
{zrllrz360,pezhou,qianluo,sunlids}@connect.hku.hk
{cenjun.cen,songyibing.syb}@alibaba-inc.com, yanchaoy@hku.hk

## Abstract

Effective generalization in robotic manipulation requires representations that capture invariant patterns of interaction across environments and tasks. We present a self-supervised framework for learning hierarchical manipulation concepts that encode these invariant patterns through cross-modal sensory correlations and multi-level temporal abstractions without requiring human annotation. Our approach combines a cross-modal correlation network that identifies persistent patterns across sensory modalities with a multi-horizon predictor that organizes representations hierarchically across temporal scales. Manipulation concepts learned through this dual structure enable policies to focus on transferable relational patterns while maintaining awareness of both immediate actions and longer-term goals. Empirical evaluation across simulated benchmarks and real-world deployments demonstrates significant performance improvements with our concept-enhanced policies. Analysis reveals that the learned concepts resemble human-interpretable manipulation primitives despite receiving no semantic supervision. This work advances both the understanding of representation learning for manipulation and provides a practical approach to enhancing robotic performance in complex scenarios. Code is available at: https://github.com/zrllrz/HiMaCon.

## 1   Introduction

Robot manipulation in diverse, unstructured environments remains a fundamental challenge. Despite advances in policy learning and architectures [4, 10, 19, 24], current approaches often fail when encountering unexpected variations or novel scenarios. As illustrated in Fig. 1, a policy trained to place cups into containers may succeed in familiar settings but fail when encountering unexpected barriers—revealing a critical generalization gap limiting real-world deployment.

We propose that addressing this challenge requires learning transferable *manipulation concepts*—hierarchical abstractions capturing fundamental manipulation patterns. These concepts connect low-level actions to high-level goals, enabling robust generalization. For example, the concept of "placing an object inside a container" encompasses invariant relational patterns that persist whether the container has barriers or not, allowing adaptation while maintaining core manipulation strategy.

To acquire these manipulation concepts, we propose a self-supervised framework that learns hierarchical latent representations without requiring labor-intensive human annotations [13, 28, 40]. Our approach operates through two complementary mechanisms: 1) *Cross-modal correlation learning* captures invariant patterns across different sensory modalities (vision, proprioception), enabling generalization across visual variations while preserving functional relationships. When placing ob-

39th Conference on Neural Information Processing Systems (NeurIPS 2025).

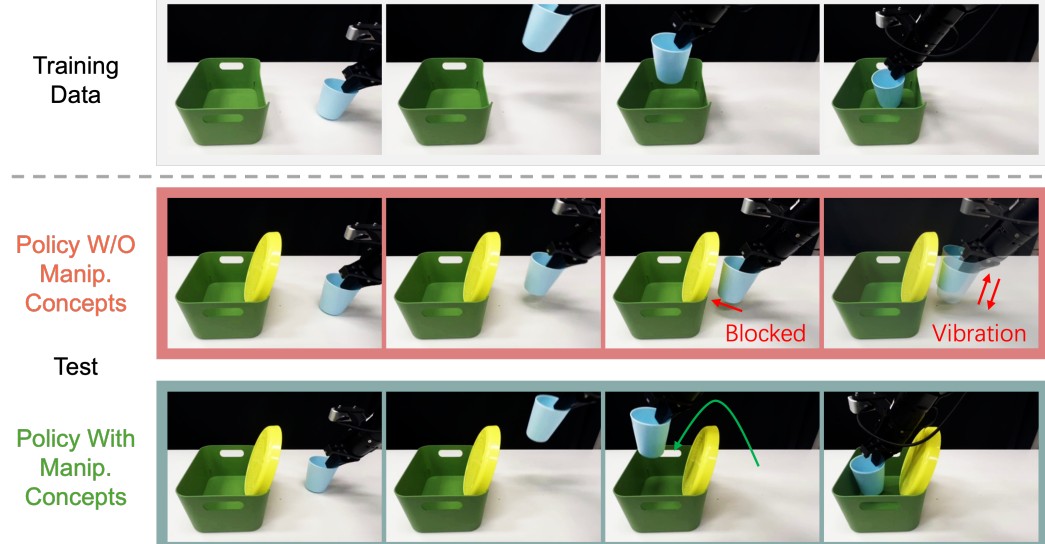

Figure 1: **Manipulation concepts enhance generalization.** Top: Training data with cups and containers without barriers. Middle: Without manipulation concepts, policies fail when encountering barriers. Bottom: With our concept enhancement, policies adapt accordingly.

jects in containers, these correlations encode the relationship between visual perception of container boundaries and proprioceptive feedback during placement, regardless of container appearance. 2) *Multi-horizon sub-goal organization* structures concepts hierarchically across temporal scales, from immediate actions (e.g., "align gripper with object") to extended sequences (e.g., "transport object to container"). This hierarchical representation enables policies to simultaneously reason about immediate actions and longer-term goals, maintaining task coherence even when specific execution paths require adaptation.

Our experiments across both simulated benchmark tasks and real-world robot deployments demonstrate that policies enhanced with these manipulation concepts consistently outperform conventional approaches, particularly in challenging scenarios requiring adaptation to novel objects, unexpected obstacles, and environmental variations (Fig. 1). The learned concepts form interpretable clusters that resemble meaningful manipulation primitives, providing insights into how robots perceive and reason about manipulation tasks.

In summary, our key contributions include: (1) a self-supervised framework that extracts structured hierarchical manipulation concepts from unlabeled multi-modal demonstrations, capturing both cross-modal correlations and multi-level temporal abstractions without human annotation; (2) an effective policy enhancement approach that integrates these concepts through joint prediction, maintaining compatibility with diverse policy architectures; and (3) comprehensive empirical evidence demonstrating significant performance improvements across diverse settings, with analyses revealing how learned concepts enable more robust generalization to novel environments.

## 2  Related Work

**Representation Learning in Robotics**  Self-supervised representation learning has emerged as a powerful approach for extracting meaningful skills [29, 32, 48] from robotic data, avoiding the need for manual annotation in methods such as [13, 28, 37, 40]. Initial efforts explored single-modality approaches for vision-based [7, 11, 51, 65, 68] and proprioception-based [26, 39, 45, 52] representation learning. Recent work integrates multiple modalities, combining vision with language [22, 36, 42, 50, 64], proprioception with vision [47, 62, 66], even richer [6, 53, 69].

These approaches typically focus on cross-modal alignment but often overlook the structured temporal patterns inherent in manipulation tasks. Parallel developments in temporal representation learning have addressed this challenge through various approaches: time-contrastive learning [27, 34, 35, 42, 65], temporal masked auto-encoding [47], and explicit modeling of state transitions across different

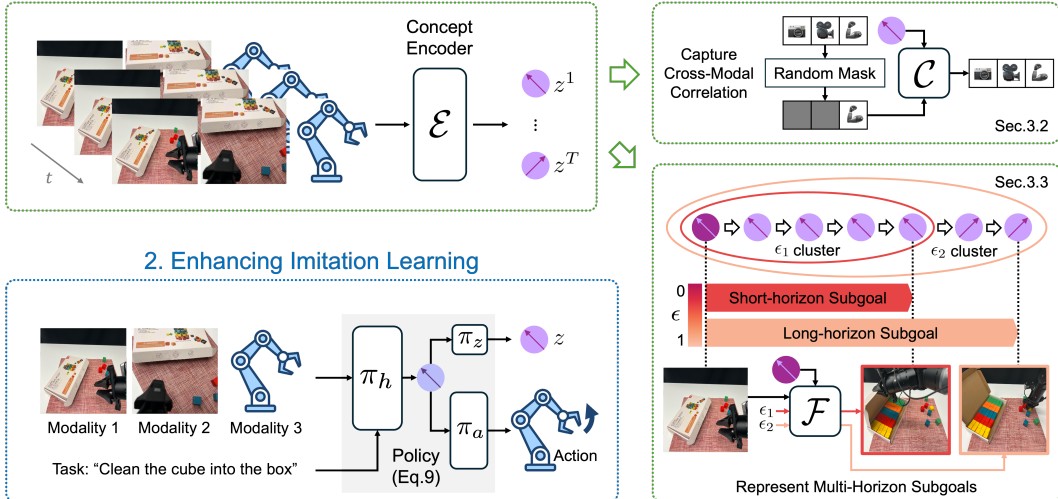

Figure 2: **The proposed self-supervised manipulation concept discovery and policy enhancement.**
*Stage 1:* The concept encoder ($\mathcal{E}$) processes multi-modal robot demonstrations to extract concept latents. These latents are refined through two objectives: (1) the Cross-Modal Correlation Network ($\mathcal{C}$) employs a mask-and-predict strategy to capture persistent patterns across sensing modalities (Sec. 3.2); (2) the Multi-Horizon Future Predictor ($\mathcal{F}$) enables concept latents to organize hierarchically into multi-horizon sub-goals based on coherence thresholds ($\epsilon$) (Sec. 3.3). *Stage 2:* The learned concepts are integrated into policy learning through a backbone network ($\pi_h$) with concept ($\pi_z$) and action ($\pi_a$) prediction heads, regularizing action generation with structured manipulation knowledge (Eq. 9).

timescales [25, 44, 54, 55]. Our work advances the field by simultaneously addressing both multi-modal integration and hierarchical temporal structures, creating representations that naturally align with manipulation sub-goals at varying time horizons while leveraging cross-modal correlational patterns that persist across different objects and contexts.

**Concept-Guided Robotic Policies**   Concept-guided approaches enhance robotic policy performance by leveraging intermediate representations that bridge perception and action. These methods generally fall into two categories. First, two-module frameworks [3, 8, 29, 62, 68, 73] employ a dual-model architecture where one component extracts high-level task concepts while another generates the corresponding actions. While effective, these approaches often require specialized architectural designs that limit their applicability across different policy classes.

Second, in contrast, joint prediction approaches [18, 20, 56, 58, 67] integrate concept guidance by training policies to simultaneously predict both concepts and actions. This creates an implicit information pathway where concept understanding regularizes action generation. Our work adopts this more flexible approach, enabling seamless integration with diverse policy architectures while maintaining the interpretability benefits of explicit concept representations.

## 3   Method

We aim to encode robotic manipulation demonstrations into latent representations that capture task-induced patterns in multi-modal sensory-motor data. These representations should naturally cluster according to functional sub-goals, providing insights into manipulation objectives and enhancing policy learning. We term these clusters *manipulation concepts*—each representing action sequences targeting specific sub-goals—and call the learning process *manipulation concept discovery*.

Our self-supervised approach works without explicit sub-goal annotations, addressing the challenge of capturing meaningful manipulation patterns without labels. We design objective functions enforcing latent representations that reflect both temporal structure and cross-modal correlations. Our approach ensures: (1) integration of modality-specific features while encoding cross-modal correlations that persist across objects and contexts; (2) hierarchical organization of sub-processes representing sub-goals across temporal horizons and enabling action prediction guided by immediate and long-term

objectives. We validate these concepts through policy performance improvements and multiple analysis methods that demonstrate correspondence with meaningful manipulation primitives.

## 3.1 Problem Setup & Manipulation Concept Encoder

Given a dataset $D = \{\tau_i\}_{i=1}^N$ of $N$ manipulation trajectories, each $\tau_i = \{(\mathbf{o}_i^t, a_i^t)\}_{t=1}^{T_i}$ contains observations $\mathbf{o}_i^t$ and actions $a_i^t$ at time $t$. For $M$ modalities, $\mathbf{o}_i^t = \{o_i^{1,t}, o_i^{2,t}, \ldots, o_i^{M,t}\}$, where $o_i^{m,t}$ is the observation of the modality $m$. We denote $\mathbf{o}_i^{S,t} = \{o_i^{m,t} \mid m \in S\}$ as observations of modalities in $S \subseteq [M] = \{1, 2, \cdots, M\}$. We treat observations of both the same sensory modes (e.g., multiple views) and different modes as distinct modalities, as they are functionally different modalities in terms of complementary information.

The *Manipulation Concept Discovery* process assigns latent representation $z_i^t \in \mathbb{R}^Z$ to each timestep $t$ of the trajectory $\tau_i$, where $z_i^t$ can be viewed as a noisy sampling of the underlying manipulation concept active at $t$. Since these representations cluster based on sub-goals, we refer to $z_i^t$ as *manipulation concept latents* or simply *manipulation concepts*. We use continuous representations for differentiability and to avoid constraints of codebook-based discrete representations (e.g., finite capacity). To learn $z_i^t$, we introduce a manipulation concept encoder $\mathcal{E}$ parameterized by $\Theta_\mathcal{E}$, which maps observation sequence $\mathbf{o}_i = \{\mathbf{o}_i^t\}_{t=1}^{T_i}$ from trajectory $\tau_i$ to concept sequence $\mathbf{z}_i = \{z_i^t\}_{t=1}^{T_i}$:

$$\mathbf{z}_i \leftarrow \mathcal{E}\left(\mathbf{o}_i; \Theta_\mathcal{E}\right) \tag{1}$$

We implement $\mathcal{E}$ using a transformer to encode temporal dependencies (details in Sec. A.1). Next, we elaborate on the training strategies optimizing cross-modal and multi-horizon temporal correlation metrics (Sec. 3.2 and 3.3).

## 3.2 Capturing Multi-Modal Correlations

To enhance the utility of multi-modal information, we propose that manipulation concepts should capture *cross-modal correlations* rather than simply aggregating features from different modalities (e.g., concatenating multi-modal signals [9, 71]). Physiological evidence suggests that concept formation often occurs when correlations across sensory modalities are high [1, 15, 38, 57, 63]. These correlations remain consistent across scenarios involving the same concept, facilitating generalization. For instance, in container opening tasks, the correlated patterns between visual lid rotation, characteristic force feedback, and audio cues persist across different container types, enabling the transfer of the "opening" concept despite variations in object appearances.

To learn manipulation concepts that capture cross-modal correlations, we propose maximizing mutual information—a metric capable of modeling diverse correlations—between observations from different modalities, conditioned on the associated manipulation concept. Specifically, we maximize the conditional mutual information over bipartitions of modality observations:

$$\max_{\mathbf{Z}} \sum_{S \subsetneq [M], S \neq \emptyset} \mathbb{I}\left(\mathbf{O}_S : \mathbf{O}_{[M]\setminus S} \mid \mathbf{Z}\right), \tag{2}$$

where $\mathbf{O}_S$ are observations from a subset of modalities, $\mathbf{O}_{[M]\setminus S}$ are observations from remaining modalities, and $\mathbf{Z}$ is the manipulation concept. We implement Eq. 2 using a computationally efficient self-supervised *mask-and-predict* approach that stochastically samples bipartitions during training. This ensures scalability despite exponentially increasing bipartition numbers while integrating cross-modal correlation learning with multi-modal information compression.

Specifically, a Cross-Modal Correlation Network $\mathcal{C}$ (CMCN) with parameters $\Theta_c$ reconstructs full-modality observations from partial observations guided by manipulation concepts. During training, we mask observations from a random subset $S$ of modalities and reconstruct all observations $\mathbf{o}_i^t$ using the unmasked subset $\mathbf{o}_i^{[M]\setminus S,t}$ and concept $z_i^t$:

$$\mathcal{L}_{\mathrm{mm}}\left(t, \tau_i\right) = \mathbb{E}_S \left\| \mathcal{C}\left(\mathbf{o}_i^{[M]\setminus S,t}, z_i^t; \Theta_c\right) - \mathbf{o}_i^t \right\|, \tag{3}$$

where $S \sim \mathrm{U}\left(2^{[M]} \setminus \{\emptyset\}\right)$ is a uniformly sampled non-empty subset of modality indices. By predicting full observations from partial inputs, we maximize the conditional mutual information in Eq. 2, forcing manipulation concepts $z_i^t$ to capture cross-modal correlations. Additionally, when all modalities are masked, reconstruction solely from $z_i^t$ ensures these representations compress and preserve essential multi-modal information (please see Sec. A.1 for more details).

## 3.3 Representing Multi-Horizon Sub-Goals

To complete tasks with hierarchical structures, manipulation concepts must encode multi-horizon sub-goal information. Physiological evidence shows human actions are hierarchically organized [17, 41], with coarse-grained goals defining overall tasks and fine-grained goals informing immediate actions. These multi-horizon sub-goals link ultimate goals with low-level actions, enabling smooth transitions while enhancing robustness.

We aim to make manipulation concepts organized to encode sub-processes across multiple temporal horizons without explicit annotations. Since concepts cluster by sub-goals, hierarchical sub-goals can emerge from these clusters at varying temporal scales. We propose that the temporal extent of a sub-process is determined by concept latent coherence within clusters, yielding a natural spectrum from short-horizon to long-horizon sub-goals. Specifically, given manipulation concept latents $\mathbf{z}_i = \{z_i^t\}_{t=1}^{T_i}$ from trajectory $\tau_i$, we quantify their similarities using spherical distance: $\text{dist}(z, u) = \frac{1}{\pi} \arccos \left\langle \frac{z}{\|z\|_2}, \frac{u}{\|u\|_2} \right\rangle$. Concepts belong to the same sub-process if their distance falls below a coherence threshold $\epsilon \in [0, 1]$. More explicitly, sub-processes are derived as:

$$
\begin{aligned}
& \text{h}\left(\mathbf{z}_i; \epsilon\right) = \left\{ [g_k, g_{k+1}) \mid k = 1, 2, \cdots, K\left(\mathbf{z}_i; \epsilon\right) \right\}, \\
& \text{where } g_1 = 1, \quad g_{k+1} = \max_g \left\{ g \mid g \in (g_k, T_i + 1] \cap \mathbb{N}^+ \wedge \forall t, t' \in [g_k, g), \text{dist}(z_i^t, z_i^{t'}) < \epsilon \right\},
\end{aligned} \tag{4}
$$

where $K(\mathbf{z}_i; \epsilon)$ is the number of clusters determined by $\epsilon$, and increasing $\epsilon$ yields sub-processes spanning from short-horizon to long-horizon. Please see Alg. 1 for more details.

Furthermore, we propose learning objectives to ensure multi-horizon sub-processes from Eq. 4 align with meaningful sub-goal completion processes. Specifically, the manipulation concept guiding each sub-process should be informative about the state achieved upon sub-task completion [5, 33, 72]. For all coherence thresholds $\epsilon$, current observation $\mathbf{O}$ and its associated concept $\mathbf{Z}$ should be informative of the terminal observation $\mathbf{O}^{\text{goal}(\epsilon)}$, characterized by minimizing the following conditional entropy:

$$
\forall \epsilon, \min_{\mathbf{Z}} \mathbb{H}\left( \mathbf{O}^{\text{goal}(\epsilon)} \mid \mathbf{O}, \mathbf{Z} \right), \tag{5}
$$

To implement Eq. 5, we train a Multi-Horizon Future Predictor $\mathcal{F}$ (MHFP) to hallucinate terminal observations of different sub-processes. For time step $t$ in trajectory $\tau_i$, the terminal observation is determined by the ending time step of the interval containing $t$:

$$
\text{g}(t; \mathbf{z}_i, \epsilon) = \min\{T_i, g_{k+1}\}, \text{ where } t \in [g_k, g_{k+1}) \in \text{h}(\mathbf{z}_i; \epsilon), \tag{6}
$$

During training, the network $\mathcal{F}$, parameterized by $\Theta_f$, predicts this terminal observation based on current observation $\mathbf{o}_i^t$, manipulation concept $z_i^t$, and coherence threshold $\epsilon$:

$$
\mathcal{L}_{\text{mh}}(t, \tau_i) = \mathbb{E}_\epsilon \left\| \mathcal{F}\left(\mathbf{o}_i^t, z_i^t, \epsilon; \Theta_f\right) - \mathbf{o}_i^{\text{g}(t; \mathbf{z}_i, \epsilon)} \right\|, \tag{7}
$$

where $\epsilon \sim \text{U}([0, 1])$ is sampled uniformly per iteration to improve efficiency by avoiding training over all $\epsilon$ values. This training process iteratively improves both latents and sub-process derivation: we compute manipulation concepts using the encoder (Eq. 1), determine sub-process boundaries, then update all networks, including $\mathcal{F}$ and the concept encoder. This improves future observation prediction and concept latents, which in turn refines sub-process derivation. By minimizing Eq. 7, $z_i^t$ is ensured to encode multi-horizon sub-goal information, indicating hierarchical transitions to terminal states under various $\epsilon$ while adjusting sub-processes by shaping concept latents for terminal state predictability. More details can be found in Sec. A.1.

***Final Objective for Manipulation Concept Discovery.*** We jointly optimize the multi-modal correlation objective (Eq. 3) and multi-horizon sub-goal prediction objective (Eq. 7) to ensure manipulation concepts generated by the encoder $\mathcal{E}$ (Eq. 1) satisfy both key properties:

$$
\mathcal{L}_{\text{z}}(t, \tau_i) = \lambda_{\text{mm}} \mathcal{L}_{\text{mm}}(t, \tau_i) + \lambda_{\text{mh}} \mathcal{L}_{\text{mh}}(t, \tau_i), \tag{8}
$$

where $\lambda_{\text{mm}}, \lambda_{\text{mh}} > 0$ balance the two loss terms.

## 3.4 Enhancing Imitation Learning with Manipulation Concepts

After learning manipulation concepts through our self-supervised framework, we address how these concepts enhance policy learning. Unlike previous approaches that learn task-specific policies

directly from demonstrations [12, 28], we propose to leverage the learned manipulation concepts as an informative representation that bridges low-level actions and high-level goals.

Specifically, with manipulation concepts $\mathbf{z}_i$ generated by encoder $\mathcal{E}$, we augment imitation learning by training policies to predict both ground-truth actions and corresponding concepts [21, 67, 70]. This approach uses concept prediction as a regularization that guides the policy to encode conceptual understanding alongside action planning:

$$h_i^t = \pi_h\left(\mathbf{o}_i^t, \ell_i; \Theta_\pi^h\right), \quad \hat{z}_i^t = \pi_z\left(h_i^t; \Theta_\pi^z\right), \quad \hat{a}_i^t = \pi_a\left(h_i^t; \Theta_\pi^a\right),$$
$$\mathcal{L}_\pi(t, \tau_i, \ell_i) = \|\hat{a}_i^t - a_i^t\| + \lambda_{\mathrm{mc}}\|\hat{z}_i^t - z_i^t\|. \tag{9}$$

The policy consists of: (1) A backbone $\pi_h$ processing task descriptions $\ell_i$ and observations $\mathbf{o}_i^t$ to produce a shared representation $h_i^t$; (2) A concept predictor $\pi_z$ mapping $h_i^t$ to predicted concepts $\hat{z}_i^t$; and (3) An action decoder $\pi_a$ mapping $h_i^t$ to predicted actions $\hat{a}_i^t$. This joint objective enforces the policy to leverage concept information encoded within $h_i^t$ while predicting actions. Even though concepts are learned task-agnostically for generalization, the policy receives task descriptions in a multi-task setting, serving as a mechanism to learn the reuse of concepts. The learning objective balances action and concept prediction using $\lambda_{\mathrm{mc}} > 0$. More details are provided in Sec. A.2.

# 4 Experiments

We evaluate our manipulation concept discovery approach through experiments addressing four key questions: (1) Do learned concepts enhance policy performance on tasks used for concept discovery, validating our strategies for encoding cross-modal correlations (Sec. 3.2) and multi-horizon sub-goals (Sec. 3.3)? (2) Can concepts learned from one task set transfer effectively to different tasks sharing underlying manipulation patterns? (3) Does our concept discovery mechanism generalize to novel tasks with decreased overlap in manipulation patterns? (4) What interpretable properties emerge in the learned concepts that explain their effectiveness for robotic manipulation? Through these investigations, we demonstrate both the immediate benefits of our approach for imitation learning and its broader applicability for transfer learning and generalization in manipulation tasks.

## 4.1 Experimental Setup

**Dataset and Environment**  Sec. 4.2 and 4.3 conduct experiments using the **LIBERO** benchmark [30], a comprehensive platform for robotic learning built on Robosuite [75]. We utilize three distinct task sets:

- **LIBERO-90**: A diverse collection of 90 manipulation tasks serving as our primary training domain for concept discovery and initial policy learning.
- **LIBERO-LONG**: 10 novel long-horizon tasks, each composed of two LIBERO-90 tasks in sequence, designed to evaluate transfer to more complex task structures.
- **LIBERO-GOAL**: 10 tasks in an entirely novel environment unseen during concept discovery, used to evaluate the generalization of learned concepts to unfamiliar contexts.

Each task includes a natural language description and 50 expert demonstrations. For multi-modal observations, we use: *Agentview vision*: 128×128 RGB third-person camera capturing the entire environment; *Eye-in-hand vision*: 128×128 RGB gripper-mounted camera; *Proprioceptive state*: 9D vector encoding gripper position, rotation, and physical states.

**Manipulation Concept Discovery Methods**  We compare our approach with several state-of-the-art concept discovery baselines (implementation details in Sec. A.3):

- **InfoCon** [31]: A VQ-VAE type of method for single-hierarchy concept discovery.
- **XSkill** [65]: Contrastive learning for manipulation skill extraction from demonstration videos.
- **DecisionNCE** [27]: Learns reward-relevant representations from demonstrations with language annotations, evaluated in two variants: using task instructions (DecisionNCE-task) and using elementary action labels (DecisionNCE-motion).
- **RPT** [47]: Temporally and modality-masked autoencoder for multi-modal sequence modeling.
- **All**: A simplified variant of our approach that predicts all modalities from concepts without modeling cross-modal correlations.

Table 1: **Evaluation of manipulation concept discovery methods across different task settings.**
Success rates (%) of ACT and Diffusion Policy (DP) models when enhanced with manipulation concepts from various discovery methods. All concept encoders were trained only on LIBERO-90, and evaluated on: original tasks (***L90-90***), novel long-horizon compositions (***L90-L***), and entirely new environments (***L90-G***). Values in parentheses show standard deviations across 4 seeds. **Bold** and underlined values indicate best and second-best results.

| ***L90-90*** | InfoCon | XSkill | RPT | All | Next | CLIP | DINOv2 | DecisionNCE task | motion | Plain | **Ours** |
|---|---|---|---|---|---|---|---|---|---|---|---|
| ACT | 66.5 (0.8) | 73.4 (0.8) | 68.8 (0.8) | 64.1 (2.0) | 68.0 (0.4) | 63.8 (0.5) | 71.9 (0.3) | 69.0 (0.1) | 66.8 (0.8) | 46.6 (1.9) | **74.8** (0.8) |
| DP | 78.2 (0.6) | 87.7 (0.6) | 84.3 (0.1) | 81.5 (0.5) | 82.6 (0.1) | 80.7 (0.9) | 79.4 (0.1) | 75.7 (0.8) | 82.7 (0.6) | 75.1 (0.6) | **89.6** (0.6) |

| ***L90-L*** | InfoCon | XSkill | RPT | All | Next | CLIP | DINOv2 | DecisionNCE task | motion | Plain | **Ours** |
|---|---|---|---|---|---|---|---|---|---|---|---|
| ACT | 55.5 (0.9) | 55.0 (1.0) | 59.0 (1.0) | 55.5 (0.9) | 55.0 (1.0) | 51.0 (1.0) | 55.0 (1.0) | 53.0 (1.0) | 49.3 (0.9) | 54.0 (0.9) | **63.0** (1.0) |
| DP | 75.0 (1.0) | 73.0 (1.0) | 61.3 (0.9) | 79.3 (0.9) | 83.0 (1.0) | 67.0 (1.0) | 63.0 (1.0) | 58.7 (0.9) | 52.7 (0.9) | 34.1 (1.1) | **89.0** (1.0) |

| ***L90-G*** | InfoCon | XSkill | RPT | All | Next | CLIP | DINOv2 | DecisionNCE task | motion | Plain | **Ours** |
|---|---|---|---|---|---|---|---|---|---|---|---|
| ACT | 67.0 (1.0) | 77.0 (1.0) | 75.0 (1.0) | 69.0 (1.0) | 71.0 (1.0) | 77.0 (1.0) | 77.3 (0.9) | 70.0 (0.9) | 75.0 (0.5) | 57.0 (1.0) | **81.0** (1.0) |
| DP | 92.7 (0.9) | 93.0 (1.0) | 91.5 (0.9) | 91.0 (1.0) | 91.3 (0.9) | 92.0 (0.9) | 91.0 (0.7) | 92.0 (0.8) | 93.0 (1.0) | 90.7 (0.9) | **95.7** (0.7) |

- **Next**: Predicts adjacent time-step observations, a common approach adopted in [7, 68].
- **CLIP** [46]: Language-aligned visual features from a pretrained foundation model.
- **DINOv2** [43]: Self-supervised visual representations without temporal modeling.
- **Plain**: Standard imitation learning without manipulation concepts.

**Policies for Concept-Enhanced Imitation Learning** To evaluate the effectiveness of our discovered manipulation concepts, we integrate them into two established imitation learning frameworks using the joint prediction approach described in Sec. 3.4:

- **ACT** [71]: A transformer-based conditional variational autoencoder that predicts action chunks.
- **Diffusion Policy (DP)** [9]: A 1D convolutional UNet that generates actions through denoising.

For both policy architectures, we add the concept prediction head ($\pi_z$ in Eq. 9) to predict manipulation concepts from the shared concept-aware representations. Implementation details appear in Sec. A.2. All experiments are reported with results aggregated across 4 random seeds.

## 4.2 Evaluating Policy Performance with Learned Manipulation Concepts

- **Performance on Original Training Tasks** We first evaluate our concept discovery method on the same tasks used for concept training. As shown in the ***L90-90*** results (Tab. 1), our approach consistently outperforms all baselines with both policy architectures. The performance gap between our method and *Next/InfoCon* demonstrates the importance of multi-hierarchical sub-goal modeling, while improvements over *All* highlight the value of explicitly capturing cross-modal correlations. Our method also surpasses *DecisionNCE* variants despite not requiring language supervision, validating the effectiveness of our self-supervised objectives.
- **Transfer to Long-Horizon Tasks** To evaluate concept transferability to more complex compositions, we apply concept encoders trained on LIBERO-90 *directly* to LIBERO-LONG demonstrations featuring novel long-horizon tasks. The ***L90-L*** results show our method maintains its performance advantage in this challenging transfer setting. This demonstrates that our approach learns manipulation concepts that effectively decompose hierarchical tasks, enabling policies to better handle novel complex task compositions requiring sequential execution of multiple sub-goals.

Table 2: **Impact of modality combinations on concept discovery performance.** Success rates (%) of ACT and DP policies using manipulation concepts discovered with different input modality combinations. All models were trained and evaluated on LIBERO-90, with specific modalities excluded (marked with "–"). A: agentview vision, H: eye-in-hand vision, P: proprioceptive state.

|  | **Ours** | – H P | A – P | A H – | – – P | – H – | A – – |
|---|---|---|---|---|---|---|---|
| ACT | 74.8±0.8 | 70.5±1.8 | 71.3±0.3 | 70.1±1.2 | 67.5±0.8 | 68.7±0.6 | 69.4±0.4 |
| DP | 89.6±0.6 | 85.8±0.2 | 85.6±0.3 | 84.3±0.5 | 84.8±0.1 | 83.7±0.1 | 85.3±0.5 |

- **Generalization to Novel Environments** We further test generalization by applying concept encoders trained on LIBERO-90 *directly* to LIBERO-GOAL demonstrations featuring unseen environments and tasks. The **L90-G** results show our method continues to outperform all baselines in this challenging scenario. This indicates our approach discovers fundamental manipulation primitives that transfer effectively across environmental variations.
- **Impact of Multi-Modal Observations** Our ablation study (Tab. 2) shows that performance consistently improves as more modalities are incorporated. The most significant drops occur when removing proprioceptive information, highlighting its importance for grounding visual observations with physical interaction states and confirming the value of our cross-modal correlation approach.

## 4.3 Analyzing Manipulation Concept Properties

**Enhanced Cross-Modal Correlation** To verify our Cross-Modal Correlation Network's effectiveness (Sec. 3.2), we measure mutual information between modalities conditioned on concept latents (Sec. A.4). Tab. 3 shows that our approach achieves higher conditional mutual information than the **All** baseline. This confirms that our mask-and-predict strategy enables the concept encoder to capture persistent cross-modal patterns that generalize across different objects and contexts, providing a robust representational basis for policies.

**Alignment with Semantic Sub-Goals** We evaluate whether our concepts align with human-understandable semantics by grouping latents from different demonstrations based on human-identified sub-goals and computing similarities between these groupings:

Table 3: **Conditional mutual information between modality pairs.** Values conditioned on concept latents from our method versus the **All** baseline that does *not* model cross-modal correlations. A: agentview, H: eye-in-hand vision, P: proprioception.

|  | Ours | All |
|---|---|---|
| $\mathbb{I}\left(\mathbf{o}_H : \mathbf{o}_A \mid \mathbf{z}\right)$ | 3.7999 | 2.0080 |
| $\mathbb{I}\left(\mathbf{o}_P : \mathbf{o}_A \mid \mathbf{z}\right)$ | 4.8319 | 3.1312 |
| $\mathbb{I}\left(\mathbf{o}_P : \mathbf{o}_H \mid \mathbf{z}\right)$ | 4.8255 | 3.1322 |

$$\langle C_i, C_j \rangle = \frac{1}{|C_i||C_j|} \sum_{z_i \in C_i} \sum_{z_j \in C_j} \left\langle \frac{z_i}{\|z_i\|_2}, \frac{z_j}{\|z_j\|_2} \right\rangle, \tag{10}$$

where $C_i$, $C_j$ represent human-identified sub-goal categories, and $z_i$, $z_j$ are latents within each category (details in Sec. C.2). As shown in Fig. 4, similarity matrices consistently show the highest values along the diagonal, demonstrating that our approach discovers concepts that exhibit clustering patterns corresponding to meaningful manipulation primitives.

**Multi-Level Hierarchical Structure** Varying the coherence threshold $\epsilon$ in Eq. 4 reveals the hierarchical organization of our learned concepts. Fig. 3 (and Sec. C.5) shows larger $\epsilon$ values identify coarse-grained phases, while smaller values capture fine-grained actions. This emergent hierarchy enables policies to simultaneously reason about immediate actions and longer-term goals without explicit hierarchical supervision, contributing to improved performance on complex sequential tasks that require coordinated execution across multiple temporal scales.

## 4.4 Real-World Validation

**Real-World Generalization Study** To study generalization capabilities, we deploy concept-enhanced policies on a Mobile ALOHA robot [16] in "cleaning cup" tasks (Fig. 5). Training data includes only simple container arrangements with consistent color pairings.

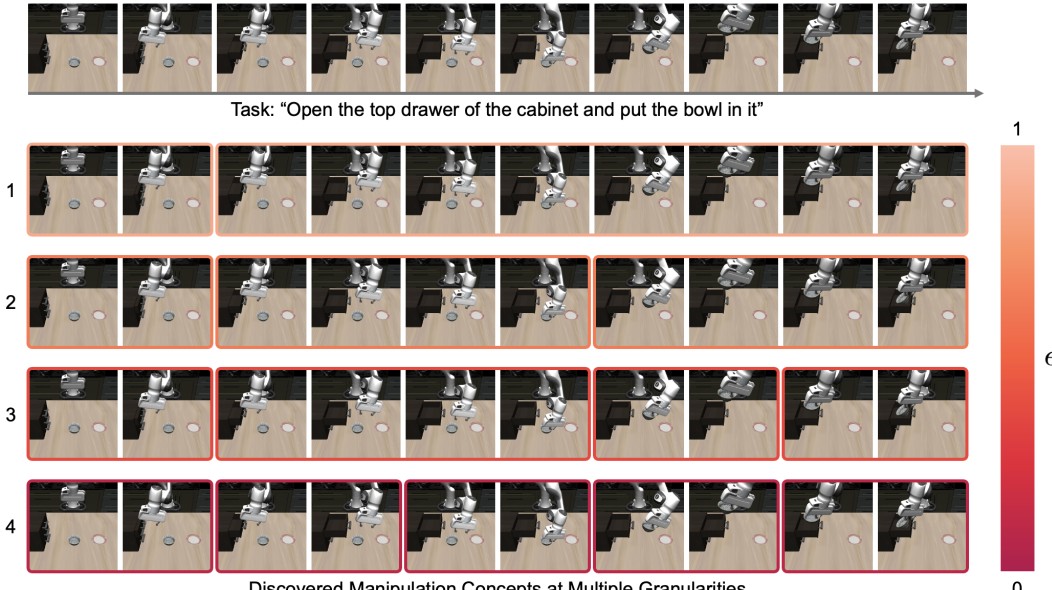

Task: "Open the top drawer of the cabinet and put the bowl in it"

Discovered Manipulation Concepts at Multiple Granularities

Figure 3: **Multi-granular task decomposition through concept latent clustering.** Visualization of sub-processes derived by clustering manipulation concept latents at different coherence thresholds ($\epsilon$) for the task "open the top drawer and put the bowl in it." Higher $\epsilon$ values (top rows) produce coarser decompositions, while lower values (bottom rows) yield finer-grained segmentation. The emergent sub-processes naturally align with semantic task components, for example, the third segment in row 2 corresponds to "put bowl in drawer," while the second segment in row 4 corresponds to "pull drawer open." This demonstrates our method's ability to discover hierarchical, human-interpretable task structures without explicit supervision.

We test on six increasingly challenging variations: **(1) Novel Placements**: Cups and containers in unseen arrangements; **(2) Color Composition**: Altered cup-container color pairings; **(3) Novel Objects**: Entirely unseen containers, cups, and plates; **(4) Obstacles**: Objects between the robot and the cups obstructing vision; **(5) Barriers**: Internal dividers within containers impeding placement; **(6) Grasping Together**: Two adjacent cups requiring simultaneous grasp.

As shown in Tab. 4, policies enhanced with our manipulation concepts consistently outperform baselines across all scenarios, with advantages in challenging conditions. We suggest that the following two mechanisms behind learned manipulation concepts improve generalization:

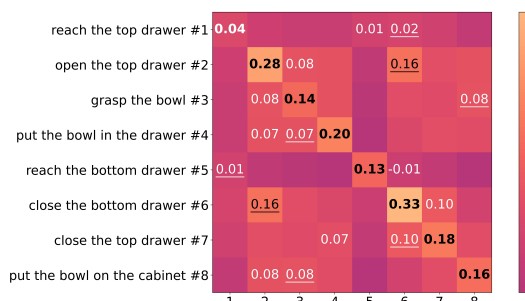

Figure 4: **Semantic alignment of learned concepts.** Cosine similarity between concept latents grouped by human-defined sub-goals. Diagonal patterns demonstrate that our approach discovers concepts that exhibit clustering patterns corresponding to meaningful manipulation primitives.

**1. Relational focus**: Concept-enhanced policies prioritize transferable relational patterns (e.g., "object inside container") over surface features. Our cross-modal correlation learning (Sec. 3.2) enables this capability by identifying patterns that remain invariant across modalities. This relational emphasis explains the stronger performance on scenarios that alter visual appearance while preserving task structure. For instance, while **Novel Placement** tests spatial variation alone, the other protocols introduce substantial visual perturbations (different colors, objects, or occlusions) that shift the appearance distribution. The consistent performance gains across these visually diverse scenarios (Tab. 4) suggest that the learned concepts successfully capture the underlying relational invariant —placing cups into containers —rather than memorizing superficial visual patterns.

**2. Hierarchical awareness**: Concept-enhanced policies exhibit more systematic failure recovery than baselines, suggesting better tracking of sub-goal completion. Baseline failures frequently exhibit

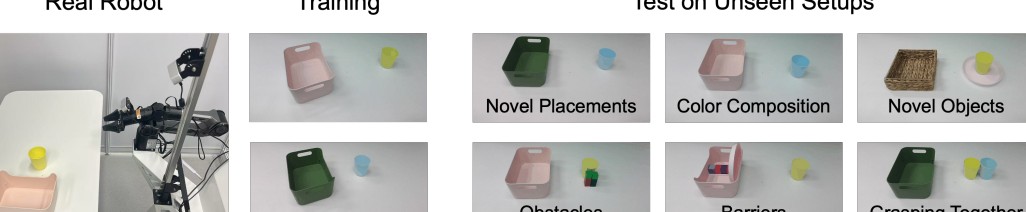

Figure 5: **Real-world generalization evaluation with Mobile ALOHA robot.** Left: Mobile ALOHA robot setup for cup cleaning tasks. Center: Training conditions with simple, consistent cup-container color pairings. Right: Six test variations with increasing complexity: novel placements, altered color combinations, unfamiliar objects, external obstacles, internal barriers, and simultaneous grasping of multiple cups. These variations test the policy's ability to generalize beyond training conditions by systematically introducing new challenges.

premature task abandonment: the robot moves toward containers without having grasped objects, or hovers near placement locations without executing placement. In contrast, when concept-enhanced policies fail initial grasp attempts, they consistently retry grasping (typically 2-3 attempts) before proceeding, demonstrating recognition of incomplete sub-goals. Although these recoveries ultimately fail due to time limits or object displacement, they reveal structured task progression rather than blind action execution.

These mechanisms may enable manipulation concepts to promote policy generalization by encoding fundamental spatial and functional relationships that remain consistent across environmental variations. Details are provided in Sec. C.6.

**Multi-Horizon Goal Prediction Visualization**   To visualize the temporal information encoded in our manipulation concepts, we examine outputs from our Multi-Horizon Goal Predictor (MHGP, $\mathcal{F}$ in Eq. 7) using the BridgeDataV2 dataset [60].

Table 4: **Real-world generalization success rates (%)** for ACT policies with and without manipulation concepts (MC). Test conditions: Placements (novel layouts), Color (new pairings), Objects (unseen items), Obstacles (external barriers), Barrier (internal dividers), and Multi-grasp (two cups simultaneously).

|        | Place | Color | Obj. | Obst. | Barr. | Multi |
|--------|-------|-------|------|-------|-------|-------|
| w/o MC | 53.3  | 46.7  | 40.0 | 20.0  | 0.0   | 0.0   |
| w/ MC  | **73.3** | **60.0** | **53.3** | **33.3** | **20.0** | **13.3** |

Fig. 7 (Sec. C.7) shows predicted goal states when conditioned on the current observation, manipulation concept, and various coherence thresholds ($\epsilon$).

The predictions capture essential task structures – such as anticipated arm trajectories and object interactions – rather than attempting pixel-perfect reconstructions. This abstraction of scene-specific details in favor of functional relationships is crucial for cross-environment generalization. Importantly, as $\epsilon$ increases, the predictions correspond to states progressively further into the future, with smaller values showing immediate next steps and larger values revealing final goal states. This demonstrates that our learned concepts encode meaningful temporal structures at multiple time horizons, enabling policies to simultaneously reason about immediate actions and longer-term objectives. Details are provided in Sec. C.7.

## 5   Discussion

We demonstrate that self-supervised discovery of hierarchical manipulation concepts significantly enhances robot policy performance across original tasks, novel compositions, and entirely new environments. Three key strengths emerge: (1) our representations naturally resemble semantically meaningful manipulation primitives without requiring explicit labels, as evidenced by diagonal clustering in similarity matrices; (2) the concepts bridge low-level actions and high-level goals through hierarchical organization, enabling reasoning at multiple temporal scales; and (3) concept-enhanced policies focus on transferable relational patterns rather than superficial features, explaining their robust generalization to scenarios with substantial distribution shifts. These findings highlight the potential of learning manipulation concepts from unlabeled multi-modal demonstrations for creating more adaptable and interpretable robotic systems. Limitations are discussed in Sec. D.

## Acknowledgments and Disclosure of Funding

This work is supported by the Early Career Scheme of the Research Grants Council (RGC) grant # 27207224, the HKU-100 Award, a donation from the Musketeers Foundation, in part by the JC STEM Lab of Robotics for Soft Materials funded by The Hong Kong Jockey Club Charities Trust, and DAMO Academy through the Alibaba Innovative Research Program.

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

# A  Implementation details

## A.1  Manipulation concept discovery (Ours)

This section details the neural network architectures and training procedures employed in our manipulation concepts discovery framework (Sec. 3) as implemented on the LIBERO benchmark.

**Manipulation Concept Encoder (Sec. 3.1)**  The manipulation concepts encoder $\mathcal{E}$ (Eq. 1) first encodes the multi-modal observations at each time step of the input demonstration into an encoded vector. It then utilizes a self-attention transformer to process the sequence of encoded vectors into a sequence of manipulation concepts. For the observation encoding process, our experiments on LIBERO incorporate two vision observations: **agent-view vision** and **eye-in-hand vision**. The original images are tensors of shape $128 \times 128 \times 3$. To enhance processing efficiency, we preprocess the images for each time step using the VAE encoder from stable diffusion [49], compressing each image into a tensor of shape $16 \times 16 \times 4$, which is then flattened into a 1024-dimensional vector. In addition to the two vision observations, we include a 9-dimensional robot state at each time step of each demonstration as the proprioceptive state observation. For these three observations at each time step, we employ three distinct 2-layer MLPs to process each observation into a feature vector of the hidden size (256) used by the subsequent transformer. The encoded features from these observations are then summed to form a 256-dimensional representation that encapsulates the sensing information from the three modalities.

$$h_{\text{av}} = \text{MLP}_{\text{av}}(I_{\text{av-compress}}) \quad h_{\text{ev}} = \text{MLP}_{\text{ev}}(I_{\text{ev-compress}}) \quad h_{\text{prop}} = \text{MLP}_{\text{prop}}(s_{\text{prop}})$$
$$h = h_{\text{av}} + h_{\text{ev}} + h_{\text{prop}}$$
(11)

Here, $I_{\text{av-compress}}$ represents the 1024-dimensional compressed **agent-view vision**, and $I_{\text{ev-compress}}$ represents the 1024-dimensional compressed **eye-in-hand vision**. $s_{\text{prop}}$ denotes the 9-dimensional proprioceptive state observation. The output of the hidden layers from the three MLPs is 1024 dimensions. The $h$ in Eq. 11 represents the encoded observation feature at each time step of a given demonstration $\tau_i$: $(h_i^1, h_i^2, \cdots, h_i^{T_i})$. The next module in $\mathcal{E}$ is a 12-layer self-attention (MHA in Eq. 12) transformer, enabling each time step to aggregate information from every other time step in the input sequence. In our implementation, we do not input the entire demonstration; instead, the transformer processes a fixed input sequence length of $T_{\text{context}} = 60$. A learnable temporal embedding, represented as a tensor of shape $60 \times 256$, is added to the input sequence to enhance temporal representation. The hidden feature dimension at each time step is 256, and each self-attention layer consists of 8 heads. Moreover, since spherical distance is utilized in Sec. 3.3, the output manipulation concepts are normalized to have a unit length with respect to the 2-norm:

$$(z_i^t, z_i^{t+1}, \cdots, z_i^{t+T_{\text{context}}-1}) \leftarrow \text{Norm}_2 \left( [\text{MHA}]_{\times 12} \left( h_i^t, h_i^{t+1}, \cdots, h_i^{t+T_{\text{context}}-1} \right) \right) \quad (12)$$

The output manipulation concept sequence in Eq. 1 represents the predicted manipulation concepts at time-steps $t, t+1, \cdots, (t+T_{\text{context}}-1)$ of the demonstration $\tau_i$. During training, demonstrations with lengths shorter than $T_{\text{context}}$ are padded to $T_{\text{context}}$ by repeating the observations from the last time-step at the end of each demonstration. During inference, when $\mathcal{E}$ is used to label the demonstrations in the original dataset, the manipulation concepts at each time step are designed to incorporate information from as many future time steps as possible. This approach aims to better capture motion pattern dynamics, aligning with prior works that generate the dynamics at the current time step based on information derived from the dynamics spanning the current to future time steps [68]. Specifically:

- For each time-step $t \leq T_i - T_{\text{context}}$, the corresponding manipulation concepts are derived when the input to Eq. 12 starts from this time-step and spans a length of $T_{\text{context}}$: $\left( h_i^t, h_i^{t+1}, \cdots, h_i^{t+T_{\text{context}}-1} \right)$.
- For each time step $t > T_i - T_{\text{context}}$, the corresponding manipulation concepts are derived when the input to Eq. 12 begins at time step $h_i^{T_i-T_{\text{context}}+1}$ and spans a length of $T_{\text{context}}$, ensuring that the final time step corresponds to the end of the demonstration: $\left( h_i^{T_i-T_{\text{context}}+1}, h_i^{T_i-T_{\text{context}}+2}, \cdots, h_i^{T_i} \right)$.
- If the original demonstration length is smaller than $T_{\text{context}}$, the manipulation concepts correspond to the input appended with repeated observations as described earlier.

However, we do not firmly believe this is the optimal approach for labeling manipulation concepts. Further exploration of inference-time strategy design is left for future work, as it is not a core focus of the manipulation concept discovery methodology presented.

**Learning Multi-Modal Features and Correlations (Sec. 3.2)** The Cross-Modal Correlation Network $\mathcal{C}$ (Eq. 3) shares a similar structure with $\mathcal{E}$ (Eq. 1). First, it includes four 2-layer MLP encoders, analogous to the three encoders in Eq. 11, with an additional encoder for processing the manipulation concepts. Each of these four MLPs outputs a hidden feature of dimension 1024, which is then reduced to a 256-dimensional encoded feature. These encoded features are summed to represent the combined information from the three observations and the manipulation concepts. Second, it incorporates a 4-layer self-attention transformer to process the sequence of features (with the same fixed length $T_{\text{context}} = 60$) produced by the four MLPs. Following this, three 3-layer MLP decoders map the transformer's output to the reconstructed observations at each time step. Unlike in Eq. 12, the transformer's output does not require normalization. Each decoder MLP has hidden layers with a dimension of 1024. As described in Eq. 3, for the three observations—**agent-view camera vision**, **eye-in-hand camera vision**, and **proprioceptive state observation**—we randomly mask these modalities, ensuring that at least one modality is masked during each iteration. The $2^3 - 1 = 7$ possible masking scenarios follow a uniform distribution, with each scenario appearing with a probability of $\frac{1}{7}$. For the sampled masks, all observations of the corresponding masked modalities in the input sequence are replaced with zero tensors. The loss is applied separately to the reconstruction of the three different observations. Specifically, L2 loss is applied to the two vision observations, while L1 loss is applied to the proprioceptive state observations. The loss weight $\lambda_{\text{mm}}$ in Eq. 8 is set to 1.0.

**Learning Multi-Hierarchical Sub-goals (Sec. 3.3)** The Multi-Horizon Future Predictor $\mathcal{F}$ (Eq. 7) shares a similar structure with $\mathcal{C}$ (Eq. 3). The key differences are:

- $\mathcal{F}$ does not require a masking strategy.
- The transformer in $\mathcal{F}$ is a 4-layer causal self-attention transformer. Causal attention is used because, in Eq. 7, the prediction is made from each current time step to certain future time steps. Therefore, for each time-step input in $\mathcal{F}$, access to information from subsequent time steps is restricted.
- To incorporate the granularity parameter $\epsilon \in [0, 1]$, we discretize the continuous range into 1000 uniform bins $\{0.000, 0.001, \ldots, 0.999\}$ and learn a corresponding VQ-VAE codebook [59] with 1000 entries, each represented as a 256-dimensional embedding vector. In each transformer block, the feed-forward layer receives the concatenation of the attention output and the embedding corresponding to the sampled $\epsilon$ value.
- The output predictions correspond to the observations at the time steps determined by the rules described in Sec. 3.3 (Eqs. 4 and 6). Still, the loss is applied separately to the reconstruction of the three types of observations. Specifically, L2 loss is used for the two vision observations, while L1 loss is applied to the proprioceptive state observations. The loss weight $\lambda_{\mathrm{mh}}$ in Eq. 8 is set to 1.0.

**Training Details**    We train the manipulation concept discovery process for 200,000 iterations with a batch size of 512. Each item in the batch is a segment of demonstration with a fixed length of $T_{\mathrm{context}} = 60$. The training process uses the AdamW optimizer with a weight decay of 0.001 and momentum parameters $\beta_1 = 0.9$ and $\beta_2 = 0.95$. The base learning rate is set to 0.001. Initially, the model is trained with a 100-iteration warmup phase, during which the learning rate increases linearly from 0.0001 to 0.001. After the warmup, the model is trained for the remaining iterations using a cosine decay schedule, gradually reducing the learning rate back to 0.0001. This training setup is compatible with GPUs such as the GeForce RTX 3090 or 4090. However, we leverage the A800 GPU for improved efficiency, completing the training process in 1.5 days.

## A.2    Enhancing Imitation Learning

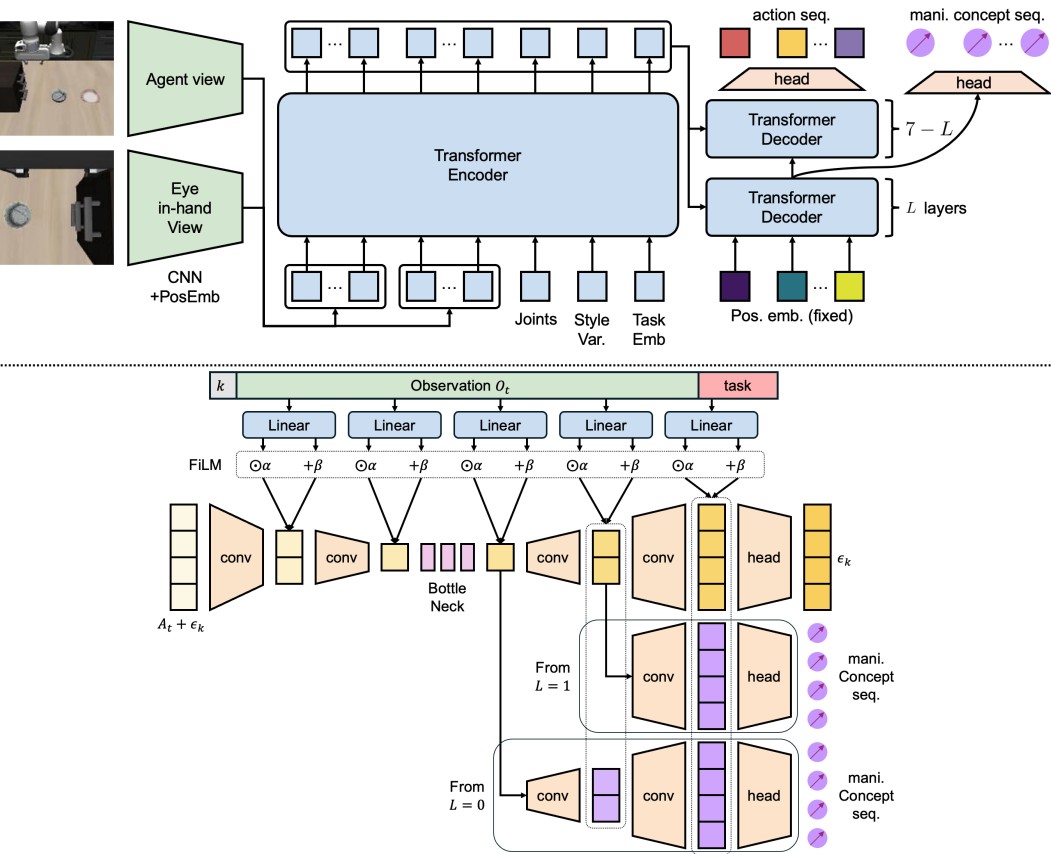

Figure 6: Upper: Enhanced ACT (decoder part); lower: Enhanced Diffusion Policy

This section introduces the neural network architectures and training details used in the Enhancing Imitation Learning process (Sec. 3.4). It focuses on modifying the original neural network policy to enable the prediction of manipulation concepts, thereby enhancing performance. Moreover, the implementation of the base policy follow [39].

**ACT** The pipeline (we focus on the CVAE decoder as it is the only modified component) is shown in the upper part of Fig. 6. Following [39], the transformer encoder in ACT's CVAE decoder is modified to incorporate task embeddings provided by CLIP. The transformer decoder in ACT's CVAE decoder is adapted to predict manipulation concepts. Specifically, the output of the $L$-th layer in the transformer decoder is processed by an additional decoding head, which is nearly identical to the one used for outputting action chunks, with the only modification being the output dimension. This decoding head outputs manipulation concept chunks corresponding to the same time steps as the action chunks, with its parameters adjusted to match the dimensionality of the manipulation concepts at each time step (256). Other training and testing settings follow [39]. Moreover, the transformer decoder in ACT's CVAE decoder, as implemented by [39], consists of 7 layers. During our experiments, we tested various combinations of $L$-th layers to determine the optimal layer for processing by the manipulation concept decoding head. Our results indicate that $L = 2$ provides slightly better performance than other configurations. We present the ablation study on $L$ and the weight $\lambda_{mc}$ in Eq. 9 for ACT on LIBERO-90 tasks, as shown in Tab. 5 [1]. However, we believe this raises an interesting and challenging direction for future work: systematically investigating the rationale and insights behind the selection of $L$, even beyond the context of our setting.

Table 5: Ablation study on the intermediate layer outputs ($L$) used as inputs to the manipulation concept decoder and the loss weight $\lambda_{mc}$ in Eq. 9 for Enhancing Imitation Learning in ACT, evaluated on LIBERO-90 tasks.

| ACT | $\lambda_{mc} = 1.0$ | $\lambda_{mc} = 0.1$ | $\lambda_{mc} = 0.01$ | $\lambda_{mc} = 0.001$ |
|---|---|---|---|---|
| $L = 2$ | **74.8±0.8** | 70.6±0.8 | 69.0±0.1 | 68.7±0.5 |
| $L = 3$ | 70.0±0.4 | 69.9±0.2 | 68.8±1.0 | 68.7±0.6 |
| $L = 4$ | 72.6±0.5 | 69.9±0.3 | 69.6±0.2 | 67.3±0.5 |

**Diffusion Policy** The pipeline is illustrated in the lower part of Fig. 6. Following [39], the convolution-based Diffusion Policy is modified to concatenate the noise level ($k$ and corresponding $\epsilon_k$) embedding, observation, and task embedding as the conditional input to the diffusion model network, using the FiLM strategy. We further introduce an additional manipulation concept decoding up-sampling module, nearly identical to the one used for outputting action chunks, with the only modification being the output dimension, to decode intermediate outputs from the corresponding up-sampling layer of the diffusion model. This decoding head can be configured to process intermediate outputs to predict manipulation concept chunks corresponding to the time steps of the predicted (noise of) action chunk outputs. The figure illustrates the cases for $L = 0$ and $L = 1$. During our experiments, we tested various combinations of $L$-th layers to identify the optimal layer for processing by the manipulation concept decoding head. Our results suggest that $L = 1$ achieves better performance than other configurations. We present the ablation study on $L$ and the weight $\lambda_{mc}$ in Eq. 9 for Diffusion Policy on LIBERO-90 tasks, as shown in Tab. 6. Similar to ACT, we believe this topic needs further systematic study to uncover deeper insights. Other training and testing settings follow [39].

Table 6: Ablation study on the intermediate layer outputs ($L$) used as inputs to the manipulation concept decoder and the loss weight $\lambda_{mc}$ in Eq. 9 for Enhancing Imitation Learning in Diffusion Policy, evaluated on LIBERO-90 tasks.

| DP | $\lambda_{mc} = 1.0$ | $\lambda_{mc} = 0.1$ | $\lambda_{mc} = 0.01$ | $\lambda_{mc} = 0.001$ |
|---|---|---|---|---|
| $L = 0$ | 83.5±0.8 | 78.9±0.4 | 78.7±0.3 | 75.6±0.6 |
| $L = 1$ | 80.0±0.4 | **89.6±0.6** | 82.0±0.2 | 79.9±0.1 |

---

[1]We provide sample rollouts (`supplementary/rollout_summary`) and .gif logs of test-time ACT and DP performance (`supplementary/rollout_video_samples_gif`) in the supplementary materials.

Our future work includes a deeper study of modification strategies for various policies to adapt to the Enhancing Imitation Learning framework, following the methodology outlined in Sec. 3.4.

### A.3 Manipulation concept discovery (Baselines)

- **InfoCon.** Based on the design of InfoCon [31], All the size of hidden features output by transformers and concept features is 256. The state encoder (also process video clips consisting of concatenated, compressed vision observations and proprioceptive states, as outlined in Sec. A.1) uses a 12-layer transformer. The state reconstructor uses a 4-layer transformer. The goal-based policy uses a 4-layer transformer. The predictor for the generative goal uses a 4-layer transformer. For hyper-network used for discriminative goals, we use 2 hidden layers in the goal function. The number of concepts is fixed, the maximum number of 30 manipulation concepts for all the tasks. We employ the AdamW optimizer, coupled with a warm-up cosine annealing scheduler same as Sec. A.1. The weight decay is always $1.0 \times 10^{-3}$. We use a batch size of $512$ during training. We train our model for 200,000 iterations with a base learning rate of $1.0 \times 10^{-3}$ on a single A800 GPU within 1.5 days.
- **XSkill.** Following the design of XSkill [65], we implement its skill discovery framework on LIBERO-90, focusing exclusively on the "robot" embodiment and the Skill Discovery component from the XSkill pipeline. To ensure comparable model capacity and support multi-modality, our implementation employs a 12-layer Transformer as the temporal skill encoder. This encoder processes video clips consisting of concatenated, compressed vision observations and proprioceptive states, as outlined in Sec. A.1, along with a trainable token to predict skill representations, which are subsequently used for skill prototype prediction. To augment the concatenated video clips containing multi-modality information, Gaussian noise with $\sigma = 1.0 \times 10^{-3}$ is applied. This unified augmentation approach accommodates the nature of proprioceptive states, as standard image augmentation techniques are not directly suitable for robotic proprioception. The training process employs a batch size of 512 and a learning rate of $1.0 \times 10^{-3}$ for 200,000 iterations on a single A800 GPU within 1.5 days.
- **DecisionNCE** We fine-tune the DecisionNCE-T model (https://github.com/2toinf/DecisionNCE) on our dataset, as it outperforms DecisionNCE-P in our analysis of the experimental results in [27]. We use two types of language annotations: (1) the original task descriptions (Decision-task), and (2) detailed subtask labels derived by decomposing each task into meaningful subprocesses (Decision-motion). To construct the latter, we manually segment each demonstration based on changes in the robot's proprioceptive state (e.g., movement direction, gripper open/close status). Segments corresponding to the same task are then assigned unified subtask labels across demonstrations, with remaining inconsistencies resolved through manual adjustment.
- **RPT.** We modify the original RPT design [47] to adapt it for our task of discovering manipulation concept latents in the LIBERO-90 setting. We employ a 16-layer self-attention transformer to process inputs consisting of 60 consecutive, interleaved agent-view and eye-in-hand vision frames. Vision inputs are compressed using a stable diffusion VAE encoder, similar to the method in Sec. A.1. The total sequence length processed by the transformer is $60 \times 3 = 180$. Each modality is mapped to a 256-dimensional embedding vector using an MLP, as defined in Eq. 11. The transformer's output is then decoded to reconstruct the original inputs using a 3-layer MLP with 1024-dimensional hidden layers. We follow the masking strategy outlined in [47] to perform temporal MAE training for the transformer. To label manipulation concept latents using the trained transformer, we extract the intermediate output of the 12th layer when the input consists of the full observation without masking. Notice that we select the output at the proprioceptive state input positions of the transformer to represent the manipulation concept latent at each time-step. The labeling process follows the procedure introduced in Sec. A.1. For training, we use a batch size of 512 and a learning rate of $1.0 \times 10^{-3}$, running for 200,000 iterations on a single A800 GPU, which completes within 3 days.
- **All.** This is an ablation version of our manipulation concept discovery method, focusing on the design for capturing multi-modal correlations (Sec. 3.2). Specifically, this baseline replaces the loss in Eq. 3 with a loss that does not use partial masking but instead always masks all modalities: $\mathcal{L}_{\text{all}}(t, \tau_i) = \left\| \mathcal{C}\left(\emptyset \middle| z_i^t; \Theta_c\right) - \mathbf{o}_i^t \right\|$. Based on our reasoning and the experiment results show in Tab. 3, we think this method may not be good at learning correlation between different modalities. Other settings follow Sec. A.1.

- **Next.** This is an ablation version of our manipulation concept discovery method, focusing on the design for representing multi-horizon subgoals (Sec. 3.3). Specifically, this baseline replaces the loss in Eq. 7 with a loss that always predicts the next adjacent time-step observation: $\mathcal{L}_{\text{next}}(t, \tau_i) = \left\| \mathcal{F}(\mathbf{o}_i^t, z_i^t; \Theta_f) - \mathbf{o}_i^{t+1} \right\|$. We observe that this setting is commonly used in recent works [7, 68], which learn representations based on adjacent time-step observations or observations separated by a fixed time horizon. We suggest that learning based on a fixed time horizon is conceptually similar to adjacent time-step settings, as the fixed time horizon can be interpreted as a unified time step. Our method differs by considering the temporal correlation across multiple variable horizons, which is also addressed by baseline methods like RPT. Other settings follow Sec. A.1.
- **CLIP.** To ensure compatibility with other baselines, which have an output dimension of 256, we select the ViT-B/32 CLIP model from the original source (https://github.com/openai/CLIP). This model outputs a 512-dimensional feature vector, the closest to 256 among the accessible CLIP models from this codebase when given an image.
- **DINOv2.** To match the output dimension of 256 used by other baselines, we select the dinov2-small DINOv2 model from the source at https://huggingface.co/facebook/dinov2-small. This model produces a 384-dimensional feature vector when given an image.

Note that DecisionNCE, CLIP, and DINOv2 baselines use only vision (and language) information for concept discovery. We preserve their original modality structure rather than adapting them to include proprioceptive states, as this would deviate from their pretraining foundations.

### A.4 Mutual information estimation

The estimation of mutual information is based on MINE [2], which uses batchwise samples drawn from a joint distribution and employs a neural network to estimate the mutual information. To extend this approach for estimating conditional mutual information (CMI), we reformulate CMI by decomposing it into mutual information terms, as shown below:

$$\mathbb{I}(X : Y \mid Z) = \mathbb{I}(X : Y) + \mathbb{I}(XY : Z) - \mathbb{I}(X : Z) - \mathbb{I}(Y : Z), \tag{13}$$

where $XY$ denotes the random variable sampled from the joint distribution of $X$ and $Y$ and is represented as the concatenation of their encoded vectors. The neural network in MINE has two layers, with the hidden layer size set to 1.5 times of the dimensions of the two random variables.

## B Pseudocode

Here we provide pseudocode for (i) Deriving subprocess from manipulation concept latents (Alg. 1). (ii) Manipulation concept disocovery training process of our method (Alg. 2).

## C More Study on Learned Manipulation Concepts

### C.1 Additional Experiments on Enhanced Imitation Learning

**Sampling Strategies** In this part, we focus on methodology for deriving hierarchical structures from learned representations (Sec. 3.3). While we adopt a threshold-based hierarchy derivation method (Eq. 4) as a proof of concept, we acknowledge that alternative derivation methodologies warrant further investigation (see Sec. D). For the threshold-based approach, we employ uniform sampling of the threshold $\epsilon$ during training. This choice ensures full coverage of all possible hierarchical structures, as we do not know a priori which threshold values might be suboptimal. To validate this design choice, we conduct an ablation study comparing different sampling strategies for $\epsilon$ in Eq. 7:

As shown in Tab. 7, uniform sampling currently achieves the best performance across both policy architectures. We hypothesize that while task-specific sampling strategies might excel on particular subsets, uniform sampling provides robust performance across diverse tasks due to its comprehensive coverage of the threshold space. Future work could explore adaptive sampling strategies tailored to specific task distributions.

Table 7: **Sampling Strategies Ablation**. We compare different sampling strategies for $\epsilon$ in Eq. 7. Manipulation concepts are learned from LIBERO-90 and applied to policy learning on LIBERO-90. We report success rates (%).

| Sampling Strategy | Description | ACT | DP |
|---|---|---|---|
| Uniform (Ours) | $\epsilon \sim \mathcal{U}(0, 1)$ | 74.8±0.8 | 89.6±0.6 |
| Sparse | $\epsilon \sim \{0.1, 0.2, \cdots, 1.0\}$ | 67.6±0.5 | 81.1±0.8 |
| Biased | $\epsilon \sim \mathcal{U}\left(\frac{1}{3}, \frac{2}{3}\right)$ | 65.6±0.7 | 78.7±0.4 |

**Learning Methodology Contribution**    We conduct an ablation study to isolate the contributions of our two core learning methodologies: Capturing Multi-Modal Correlations (Sec. 3.2) and Representing Multi-Horizon Sub-Goals (Sec. 3.3). Tab. 8 compares three configurations: (1) *Cross-modal only*: learning with only cross-modal alignment objectives in Eq. 3, (2) *Multi-horizon only*: learning with multi-horizon sub-goal prediction in Eq. 7 but without cross-modal alignment, and (3) *Full method*: combining both cross-modal alignment and multi-horizon prediction.

Table 8: **Methodology Contribution Ablation**.  We evaluate the contribution of each learning component by training manipulation concepts on LIBERO-90 and applying them to policy learning on LIBERO-90. We report success rates (%).

| Method | ACT | DP |
|---|---|---|
| Cross-modal only | 69.1±0.6 | 82.8±1.0 |
| Multi-horizon only | 71.6±0.4 | 80.5±0.5 |
| Ours (Full method) | 74.8±0.8 | 89.6±0.6 |

The results in Tab. 8 reveal that both components make substantial and complementary contributions to performance. We attribute this synergy to the distinct roles of each component: cross-modal alignment grounds the understanding of correlations across different modalities, while multi-horizon prediction captures hierarchical temporal structure. Together, they enable the learning of manipulation concepts that are both correlationally coherent and temporally structured, leading to more robust policy learning.

**Data Constraint Experiments**    We evaluate whether manipulation concepts can help mitigate the challenges of imitation learning under limited data. Specifically, we vary the amount of data available for training both the manipulation concept encoder (Eq. 1) and the enhanced imitation learning framework (Sec. 3.4) to assess their impact on policy success rates. We conduct experiments on LIBERO-90 tasks using the diffusion policy. As shown in Tab. 9, incorporating manipulation concepts consistently improves policy performance compared to settings without them, even under restricted data conditions. This demonstrates that learning and leveraging manipulation concepts can make imitation learning more data-efficient and effective.

Table 9: **Performance under data constraints.** Success rates of diffusion policies with and without manipulation concept enhancement, evaluated on LIBERO-90 (***L90-90***). In each setting, the number of demonstrations per task available for training both the manipulation concept encoder and the policy is limited as indicated.

| | 50 demos/task | 25 demos/task | 10 demos/task |
|---|---|---|---|
| Ours | 89.6 ± 0.6 | 77.6 ± 0.5 | 61.2 ± 1.1 |
| Plain | 75.1 ± 0.6 | 70.1 ± 0.3 | 59.1 ± 0.9 |

**Distance Metric**    We conduct an ablation study comparing spherical distance and cosine distance $\frac{1-\cos(\cdot)}{2}$ for $\text{dist}(\cdot, \cdot)$ in Eq. 4. Tab. 10 reports the performance when concepts are learned and applied

to LIBERO-90 tasks. Further investigation into distance-threshold-based subprocess derivation methods represents a promising direction for future work.

Table 10: Ablation study on distance metrics for concept learning on LIBERO-90. Spherical distance consistently outperforms cosine distance across both baseline methods.

|  | Cosine Distance | Spherical Distance (Ours) |
|---|---|---|
| ACT | 67.8±0.5 | 74.8±0.8 |
| DP | 82.0±0.4 | 89.6±0.6 |

**Sub-process Derivation**  We conduct an ablation study comparing two approaches for constraining manipulation concept latents within each sub-process in Eq. 4. Our proposed method enforces proximity among all concept latents throughout the sub-process ("Sequential Constraint"), while the baseline only constrains the distance between the initial and final concept latents ("Endpoint Constraint"). We evaluate both approaches on LIBERO-90, where concept discovery and policy enhancement are performed. Tab. 11 reports the task success rates when integrating the learned manipulation concepts with different policy architectures.

Table 11: Ablation study on sub-process derivation constraints. We compare enforcing proximity among all manipulation concept latents within each sub-process (Sequential Constraint) versus constraining only the initial and final latents (Endpoint Constraint). Results show average success rates (%) with standard errors across LIBERO-90 tasks.

|  | Sequential Constraint | Endpoint Constraint |
|---|---|---|
| ACT | 74.8±0.8 | 68.4±0.8 |
| DP | 89.6±0.6 | 79.8±0.5 |

**Future Prediction Strategy**  Apart from the different sub-goal determination strategies we compared (**Next** and **InfoCon** in Sec. 4.1), we evaluate two additional future prediction strategies.

- **Next-n.** Unlike our sub-process derivation strategy (Eq. 4), this baseline encodes future observations at varying time horizons by randomly sampling a future timestep. Specifically: $\mathcal{L}_{\text{next-n}}(t, \tau_i) = \mathbb{E}_{n \sim \mathrm{U}\{1,2,\cdots,T_i-t\}} \left\| \mathcal{F}(\mathbf{o}_i^t, z_i^t, n; \Theta_f) - \mathbf{o}_i^{t+n} \right\|$.
- **Next-random.** This strategy builds upon **Next-n** but differs in how future targets are selected. We first randomly segment training demonstrations into sub-processes for concept discovery. Then, for a state at time-step $t$, the prediction target is randomly selected from among the end-states of subsequent sub-processes. For example, if a demonstration is segmented into 5 sub-processes and time-step $t$ is in the 2nd sub-process, the model will randomly predict one of the end-states from the 2nd, 3rd, 4th, or 5th sub-processes during concept discovery learning.

We evaluated diffusion policies enhanced by these strategies, with results presented in Tab. 12. The data demonstrates that our manipulation concepts yield better policy enhancement compared to the alternative strategies. This highlights the importance of carefully designing which future observations to predict and validates the effectiveness of our self-supervised sub-goal derivation and learning method. Specifically, the performance decrease observed with **Next-n** and **Next-random**, despite their consideration of multi-horizon futures, likely stems from the fact that not all future states effectively represent sub-goal completion. Intermediate movement states may be reached through multiple alternative trajectories that ultimately achieve the same sub-goal, thus providing limited information about the underlying task structure.

**Usage of Manipulation Concept Encoder**  We investigate two strategies for leveraging the manipulation concept encoder from Eq. 1 in downstream policy learning. The encoder serves as an intermediate module that extracts manipulation concept representations from demonstrations. We compare the following approaches: (1) **Direct Conditioning**: The trained encoder directly processes current observations to generate manipulation concepts, which are then concatenated with observations as additional input features to the policy network. (2) **Joint Prediction (Ours)**: The policy

Table 12: **Comparison of Additional Future Prediction Strategies.** Success rates of diffusion policies enhanced with manipulation concepts discovered using our method versus two alternative future prediction strategies on the LIBERO-90 benchmark.

| L90-90 | Ours | Next-n | Next-random |
|--------|------|--------|-------------|
| DP | 89.6 ± 0.6 | 83.0±0.3 | 82.8 ± 0.4 |

network is trained to jointly predict both future actions and future manipulation concepts from current observations, as described in Sec. 3.4. Tab. 13 presents the comparative results across two policy architectures.

Table 13: **Comparison of Manipulation Concept Usage Strategies.**

| Policy | Direct Conditioning | Joint Prediction (Ours) |
|--------|---------------------|-------------------------|
| ACT | 71.1±0.4 | 74.8±0.8 |
| DP | 79.3±0.9 | 89.6±0.6 |

The performance gap stems from a temporal alignment mismatch between concept representations and action predictions. In **Direct Conditioning**, the encoder extracts concepts from current or past observations, creating a temporal lag: the policy receives historical concept information when planning future actions. In contrast, Joint Prediction enforces temporal coherence by training the policy to predict future manipulation concepts alongside future actions, ensuring that the predicted concepts align temporally with the planned action sequence.

This temporal alignment is critical in multi-phase manipulation tasks. For example, consider a pick-and-place scenario: immediately after grasping an object, the current observation encodes grasping-related dynamics. However, to execute the subsequent placement action, the policy requires placement-relevant information. Joint Prediction learns to anticipate these future task-phase concepts, providing the policy with forward-looking contextual information. Direct Conditioning, by contrast, conditions the policy on backward-looking grasping concepts that offer limited guidance for placement planning.

While our results demonstrate the advantages of temporal alignment through joint prediction, we acknowledge that direct conditioning on historical concepts may benefit tasks requiring explicit long-horizon memory or reactive behaviors based on past states [14]. Future work will explore hybrid architectures that combine both strategies.

## C.2 Alignment with Semantic Sub-Goals

We evaluate whether the manipulation concept latents learned by our method resemble human-interpretable semantics. Specifically, we assess whether latents assigned to time steps of demonstrations (Sec. 3.1) exhibit higher pairwise similarity when those steps belong to sub-processes pursuing the same human-defined sub-goal.

To analyze the learned representations, we first group manipulation concept latents according to human-annotated sub-goals. For instance, in the task "open the top drawer", latents from time steps where the robot reaches for the top drawer handle are categorized as "reach the top drawer". Latents from other demonstrations and tasks involving identical processes (reaching the top drawer) are placed in the same category. We then quantify the similarity between two categories by calculating the average cosine similarity between their respective latents, as defined in Eq. 10.

Fig. 9 shows results from analyzing demonstrations from three tasks:

- Task #1: Open the top drawer of the cabinet and put the bowl in it;
- Task #2: Close the bottom drawer of the cabinet and open the top drawer;
- Task #3: Close the top drawer of the cabinet and put the black bowl on top.

We selected these tasks because they clearly demonstrate overlapping subgoals across different tasks (e.g., Task #1 and Task #2 both include "opening the top drawer"). This enables testing whether the

latents capture similar subgoal semantics across different tasks—an essential capability for cross-task learning efficiency (Sec. 1). Manipulation concept latents are grouped based on human-defined sub-goals, with similarities between category pairs visualized as heatmaps. Three heatmaps are presented, each using a different granularity of sub-goal annotation:

1. Top-1st heatmap: Omits task-specific distinctions, merging similar manipulation processes across tasks into the same category
2. Top-2nd heatmap: Further merges similar manipulation processes, disregarding distinctions like "top drawer" versus "bottom drawer"
3. Top-3rd heatmap: Consolidates manipulation processes further, treating actions like bowl transitions as the same concept regardless of context

In each heatmap, the entry at position $(i, j)$ represents the average similarity ($\times 10.0$) between categories $i$ and $j$. For readability, only the top three similarity values in each row are displayed.

We emphasize that testing semantic capture at different "description granularity levels" is important because semantics naturally exist at multiple levels of abstraction, from highly specific details to broadly generalizable patterns. Finer-grained descriptions provide more precise details but limited generalization, while coarser-grained descriptions capture more general features applicable across diverse scenarios. For example, the general instruction "close the drawer" applies broadly to subprocesses in both Task #2 and Task #3, whereas the more specific "close the top drawer" incorporates spatial features that make it applicable in Task #3 but not in Task #2. Through this multi-granularity analysis, we evaluate whether our manipulation concept latents successfully capture both fine-grained semantics needed for specific scenarios and coarse-grained semantics that enable transfer across more scenarios.

What we observe is that the highest similarity values consistently appear along the diagonal in each heatmap in Fig. 9, so concept latents from the same category show higher similarity compared with different categories. This indicates that the learned latent clusters resemble clusters derived from human-interpretable sub-goal classifications, suggesting that our model captures meaningful semantic structure in the manipulation processes. Moreover, the patterns observed across the three heatmaps with different description granularities reveal that the latents encode semantics at multiple levels of abstraction. They capture both generalizable semantics applicable across tasks and scenes, while simultaneously preserving fine-grained scene-specific details.

Furthermore, Fig. 10(b) provides a t-SNE visualization of manipulation concept latents from all 90 tasks in LIBERO-90. For each task, latents ($z_i^t$) were extracted at every time step of demonstrations. In the plot, latents are color-coded by their originating tasks. We observe that clusters often contain latents from diverse tasks, as indicated by the mixed colors in each cluster. This further supports our finding that the learned latents generalize across tasks and capture shared semantic structures.[2]

## C.3 Motion Study

We evaluate whether the learned manipulation concept latents capture the robot's motion. Using Eq. 10, we calculate the average similarity ($\times 100.0$) between movements based on manipulation concept latents corresponding to specific gripper actions. Specifically, we collect latents for the following movements from task demonstrations in LIBERO-90:

1. Forward-backward motion: Latents for time-steps where the robot moves forward, backward, or remains still along the forward-backward axis.
2. Left-right motion: Latents for time-steps where the robot moves left, right, or remains still along the left-right axis.
3. Up-down motion: Latents for time-steps where the robot moves up, down, or remains still along the up-down axis.
4. Gripper state: Latents for time-steps where the gripper opens or closes.

---

[2]It should be noted that t-SNE performs extreme dimensionality reduction, so these clusters may not perfectly reflect similarity in the high-dimensional space. This visualization should therefore be considered as supplementary evidence.

Movements with velocities below 20% of the maximum observed velocity are classified as "still". Using these collected latents, we generate heatmaps (similar to Fig. 9) to visualize the average cosine similarity across different movement directions and gripper states (Fig. 10(a)).

The heatmaps reveal that the highest cosine similarity values often appear along the diagonal. This demonstrates that latents corresponding to the same motion patterns exhibit greater similarity to each other than to those from different motion patterns, indicating that the latents effectively capture different movement directions and gripper states. However, we observe that forward-backward motion is captured with lower accuracy compared to other dimensions. We hypothesize that incorporating additional 3D-informative modalities, such as depth maps, beyond the current proprioceptive states could improve the representation of motion along the forward-backward axis. We leave the exploration of such modality incorporation to future work.

### C.4   Diversity & Discrimination Study

We analyze the diversity and discriminability of learned manipulation concepts by comparing concept latents from our method (Sec. 3) and the baselines in *Manipulation Concept Discovery Baselines* (Sec. 4.1). Specifically, we cluster latents from these methods and examine the number of clusters under varying granularities. The number of clusters reflects concept diversity: more clusters indicate a wider variety of concepts. Clustering granularity determines whether clusters are fine-grained (fine granularity) or general (coarse granularity). Additionally, small granularity perturbations test discriminability, as less discriminative latents lead to significant clustering changes under small granularity variations. For each method, We collect manipulation concept latents from 90 LIBERO-90 tasks (one demonstration per task) and use DBSCAN to cluster them while varying the density parameter `Eps`, which controls clustering granularity. Fig. 11 shows the cluster counts across different `Eps` values. From Fig. 11, our manipulation concept discovery method (**Ours**) demonstrates two key advantages: 1) At higher granularities (`Eps` > 0.2), **Ours** maintains a **higher number of clusters**. 2) The **decline in cluster count is relatively smooth and gradual**, showing stability under small `Eps` changes. These results highlight the superior diversity and discriminability of our manipulation concept discovery method.

### C.5   Multi-Level Hierarchical Structure

In Fig. 3, we present a visualization example of the **Multi-Level Hierarchical Structure** described in Sec. 4.3. Additional visualization results are available in the supplementary materials under the directory `supplementary/vis_multi_h`.

### C.6   Real Robot Experiments Details

*Training Data*. As shown in Fig. 5, the training data for the "cleaning cup" task consists of demonstrations using mobile ALOHA [16] to place the cup from the table into the container. Each demonstration features a scene containing exactly one cup and one container. There are two pairings of color combinations: blue cups with green containers and yellow cups with pink containers. For each pairing, we collect 27 demonstrations with varied spatial arrangements.

*Evaluation Setting*. For evaluation, we test our model on six scenarios that introduce variations absent from the training data:

- **Novel Placements.** Objects maintain the same color pairings as in training but appear in previously unseen spatial arrangements.
- **Color Composition.** We rearrange color pairings (blue cups with pink containers and yellow cups with green containers) to test generalization across color combinations.
- **Novel Objects.** We introduce unseen objects, such as bamboo-woven containers, pink cups not present in training, or cups initially placed on plates rather than directly on the table.
- **Obstacles.** We position obstacles in front of cups to challenge visual perception.
- **Barriers.** We place a plate inside the container, requiring the robot to lift the cup high enough to clear this barrier when depositing it.
- **Grasping Together.** We position two cups adjacent to one another, requiring the robot to grasp both simultaneously at their contact point and deposit them together in the container.

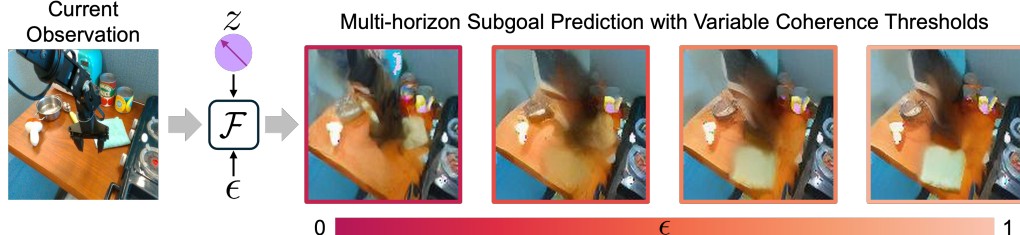

Figure 7: **Multi-horizon goal prediction with learned manipulation concepts.** Visualization of future states predicted by our Multi-Horizon Goal Predictor (MHGP, Eq. 7) when conditioned on the current observation, a manipulation concept latent ($z$), and varying coherence thresholds ($\epsilon$). From left to right, as $\epsilon$ increases from 0 to 1, predictions extend progressively further into the future, demonstrating how our manipulation concepts encode temporal abstraction at multiple horizons. Note that predictions capture essential functional relationships (robot-object interactions) rather than pixel-perfect reconstructions, facilitating generalization across environments.

*Manipulation Concept Discovery*. The model architecture and hyperparameter configuration for manipulation concept discovery follow the methodology described in Sec. A.1. Since the dataset is relatively small, we adapt smaller transformers: a 4-layer concept encoder ($\mathcal{E}$, Eq. 1), a 4-layer Cross-Modal Correlation Network ($\mathcal{C}$, Eq. 3), and a 4-layer Multi-Horizon Future Predictor ($\mathcal{F}$, Eq. 7). For data collected using mobile ALOHA [16], we incorporate the following modalities: three $640 \times 480$ resolution cameras (left-gripper, right-gripper, and upper-gripper) and 42-dimensional proprioception states (comprising 14-dimensional joint torque, position, and velocity measurements). All image data undergoes preprocessing as detailed in Sec. A.1.

*Enhancing Imitation Learning*. Please refer to **ACT** section in Sec. A.2.

## C.7 Multi-Horizon Goal Prediction Visualization

We provide visualization results of the **Multi-Horizon Goal Prediction Visualization** (Sec. 4.4) in Fig. 7 and supplementary materials under the directory `supplementary/prediction`. Below are the details of the experiments:

**Dataset.** For our experiments, we utilized the BridgeDataV2 dataset [60]. Since multi-view data is not universally available across all demonstrations, we selected two specific modalities: the robot's proprioceptive states (7DoF) and the third-person camera view. The camera images were preprocessed to $128 \times 128$ resolution following the procedure outlined in Sec. A.2.

**Manipulation Concept Discovery.** We implemented the model architecture and hyperparameter configuration as detailed in Sec. A.1, adapting it specifically to operate with the two modalities described in the **Dataset** section above.

## C.8 Preliminary VLA Integration

We present a preliminary exploration of integrating manipulation concepts with vision-language-action models (VLAs). We build upon OpenVLA-OFT [23], which fine-tunes OpenVLA using pretrained parameters and a novel action adapter for downstream tasks. The action adapter processes hidden layer features from the original pretrained VLA model. Following this architecture, we introduce an additional "concept adapter" that implements the method described in Sec. 3.4, enabling the integration of manipulation concepts into the VLA.

To evaluate the data efficiency gains from manipulation concepts, we fine-tune the enhanced VLA on **50% of the training data** used for LIBERO-10 tasks in the original OpenVLA-OFT study [23]. We compare fine-tuning performance with and without manipulation concept integration. Fig. 8 presents the results, where the x-axis indicates training epochs and the y-axis shows success rates for checkpoints at each epoch. The solid lines labeled "best" represent the highest success rate achieved up to that epoch.

The results demonstrate that manipulation concepts improve data utilization. With only **half the training data**, the concept-enhanced approach consistently achieves higher success rates throughout training. Notably, the original OpenVLA-OFT achieved 94.5% success with the full dataset [23], while our concept-enhanced model with **half the data** reaches comparable performance levels, indicating substantially improved data efficiency.

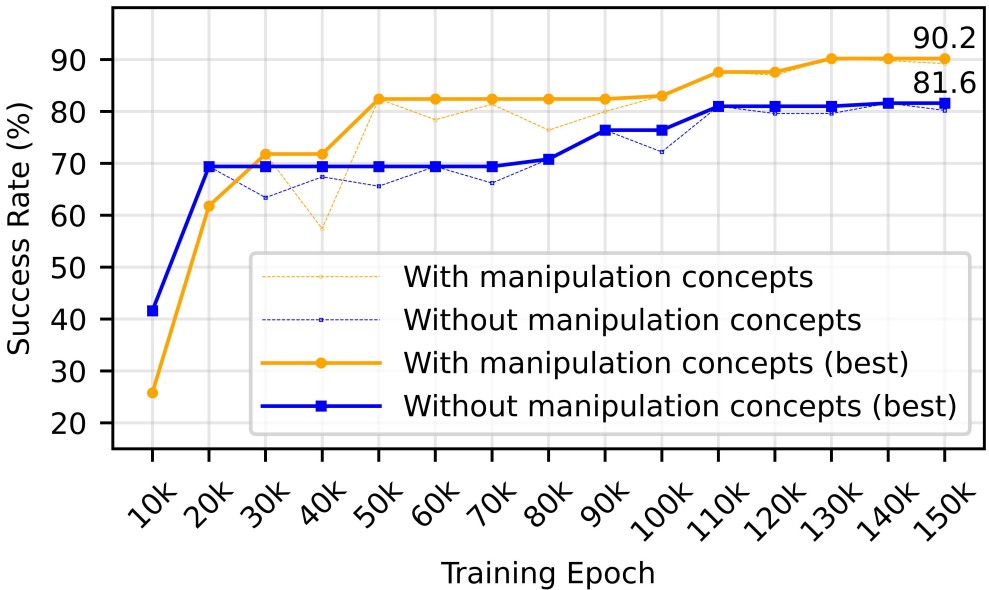

Figure 8: Data efficiency comparison on LIBERO-10 tasks with **50% training data**. Solid lines show best performance up to each epoch for models with and without manipulation concepts.

We hypothesize that this improvement stems from HiMaCon's ability to capture manipulation dynamics at multiple abstraction levels. The learned concepts provide explicit intermediate representations that bridge high-level task instructions and low-level control actions, thereby reducing the learning burden on VLAs by supplying structured manipulation knowledge rather than requiring learning of complex sensorimotor patterns from scratch. Further investigation of this integration will be pursued in future work.

## D  Limitations & Future works

**Further Exploration of multi-modality.** We propose enhancing robotic data collection with richer modalities and studying how these modalities can derive more effective manipulation concepts. While current robotics research primarily focuses on visual information, human manipulation relies on multiple sensory inputs, particularly tactile feedback to complement vision. This is especially crucial for robotic systems with limited tactile capabilities. Future work should investigate which modalities contribute most significantly to performance improvements and how to fully leverage their potential.

**Further Exploration of multi-horizon sub-goal.** Our work proposes methods to derive sub-processes for achieving sub-goals across multiple horizons, though several improvements remain possible. Current methods inadequately capture relationships between different values of $\epsilon$ in Eq. 4, failing to reflect the natural tree structure of hierarchical sub-goals. Future research could explicitly derive tree structures [61, 74] where long-horizon sub-goals serve as parent nodes to short-horizon child nodes. Additionally, our cosine similarity approach for determining sub-goal correspondence could be refined with more sophisticated metrics.

**Scaling up.** Computational constraints have limited our exploration of how our method scales with larger datasets. We plan to leverage pretrained multi-modal foundation models, adopting structures inspired by [7] and extending pretraining beyond robotics data as in [68]. We also aim to further investigate whether our manipulation concepts can enhance advanced policies like Vision-Language-Action models [4, 19, 23, 24].

---

**Algorithm 1** Derive Subprocess $h(\mathbf{z}_i; \epsilon)$

---

**Input:** manipulation concept vectors $\mathbf{z}_i = \{z_i^t\}_{t=1}^{T_i}$, coherence parameter $\epsilon \in [0,1]$.
**Initialize:** $End = []$, $g_b = 1$
**while** $g_b \leq T_i$ **do**
   $g_e = g_b + 1$
   **while** $true$ **do**
     **if** $\exists u \in [g_b, g_e)$, s.t. $\mathrm{dist}(z_i^u, z_i^{g_e}) \geq \epsilon$ **or** $g_e > T_i$ **then**
       **break**
     **end if**
     $g_e = g_e + 1$
   **end while**
   $End.append\,([g_b, g_e])$
   $g_b = g_e$
**end while**
**Return** $End$

---

---

**Algorithm 2** Manipulation Concept Discovery Training (one demonstration per batch)

---

**Input:** demonstrations $\tau_i \in D$, where $\tau_i = \{(o_i^{1,t}, o_i^{2,t}, ..., o_i^{M,t}, a_i^t)\}_{t=1}^{T_i}$
**Initialize:** Manipulation concept assignment encoder $\mathcal{E}(\cdot; \Theta_{\mathcal{E}})$
**Initialize:** Modality Correlation Learner $\mathcal{C}(\cdot; \Theta_c)$, Subgoal Learner $\mathcal{F}(\cdot; \Theta_f)$
**while** $true$ **do**
   **for** $\tau_i$ **in** $D$ **do**
     $(z_i^1, \cdots, z_i^{T_i}) \leftarrow \mathcal{E}\left((o_i^{1,1}, ..., o_i^{M,1}), (o_i^{1,2}, ..., o_i^{M,2}), \cdots, (o_i^{1,T_i}, ..., o_i^{M,T_i}); \Theta_{\mathcal{E}}\right)$
     **while** True **do**
       Randomly generate a tuple $(m_1, m_2, \ldots, m_M)$, where $m_i \in \{0, 1\}$
       **if** $\sum_{i=1}^{M} m_i < M$ **then**
         **break**
       **end if**
     **end while**
     $(\hat{o}_i^{1,t}, \cdots, \hat{o}_i^{M,t})_{t=1}^{T_i} \leftarrow \mathcal{C}\left((o_i^{1,t} \cdot m_1, o_i^{2,t} \cdot m_2, ..., o_i^{M,t} \cdot m_M, z_i^t)_{t=1}^{T_i}; \Theta_c\right)$
     $\mathcal{L}_{\mathrm{mm}} = \sum_{t=1}^{T_i} \sum_{m=1}^{M} \left\| \hat{o}_i^{m,t} - o_i^{m,t} \right\|$
     $\epsilon \sim \mathrm{U}([0,1])$
     $End = h(z_i^1, \cdots, z_i^{T_i}; \epsilon)$ {Alg. 1}
     **for** $t = 1$ **to** $T_i$ **do**
       $\mathbf{g}_t = \min\left(\left\{g_e \mid [g_b, g_e] \in End, g_e > t\right\} \cup \{T_i\}\right)$
     **end for**
     $(\overline{o}_i^{1,t}, \cdots, \overline{o}_i^{M,t})_{t=1}^{T_i} \leftarrow \mathcal{F}\left((o_i^{1,t}, o_i^{2,t}, ..., o_i^{M,t}, z_i^t, \epsilon)_{t=1}^{T_i}; \Theta_f\right)$
     $\mathcal{L}_{\mathrm{mh}} = \sum_{t=1}^{T_i} \sum_{m=1}^{M} \left\| \overline{o}_i^{m,t} - o_i^{m,\mathbf{g}_t} \right\|$
   **end for**
**end while**

---

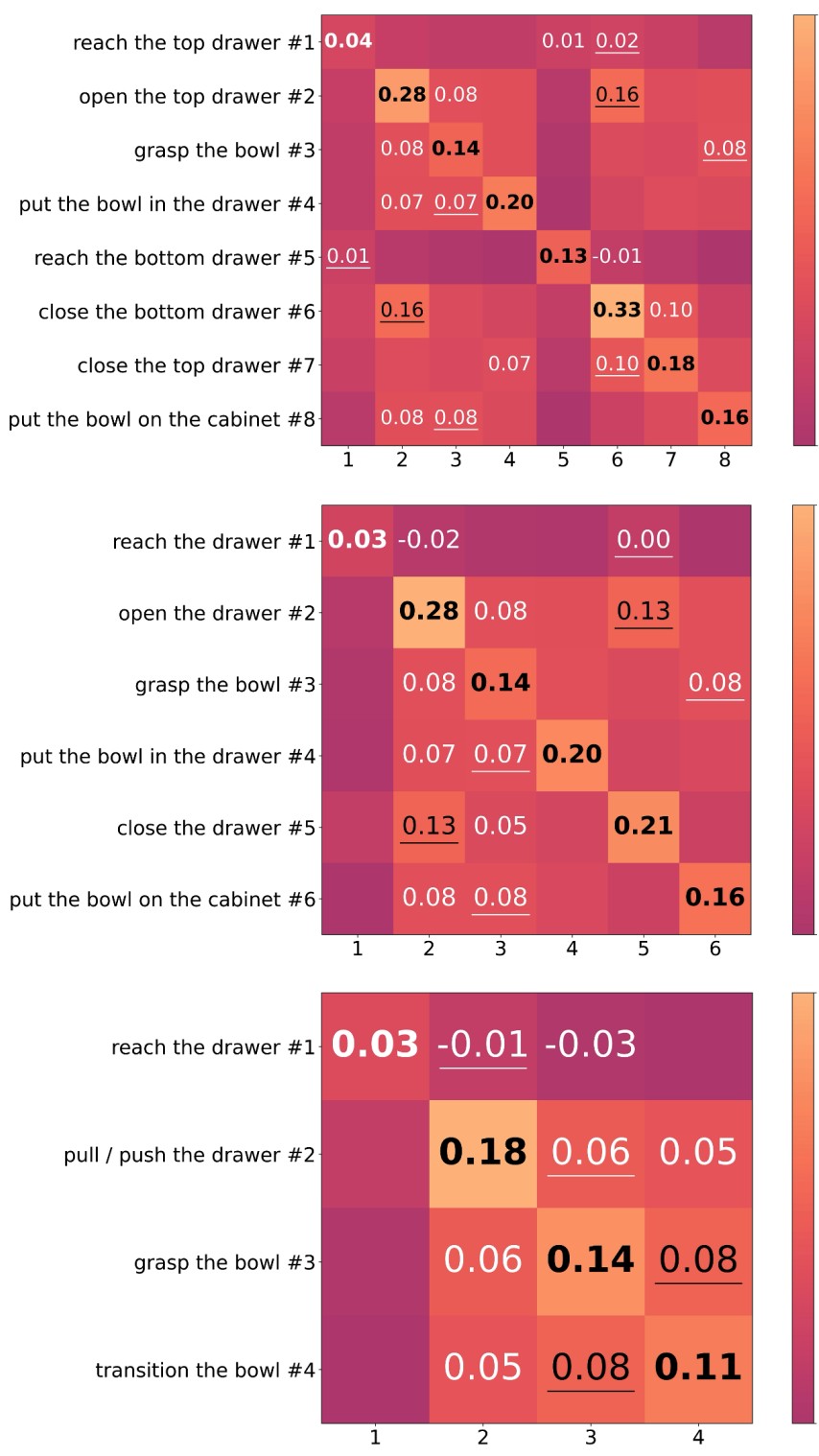

Figure 9: Average cosine similarity between pairs of sub-goal categories (defined by human semantics) computed using manipulation concept latents learned by our method (Sec.3). In each heatmap, the value at the $i$-th row and $j$-th column represents the average cosine similarity between latent vectors from the $i$-th and $j$-th categories. Three levels of labeling are provided across the heatmaps; please refer to Sec. C.2 for details.

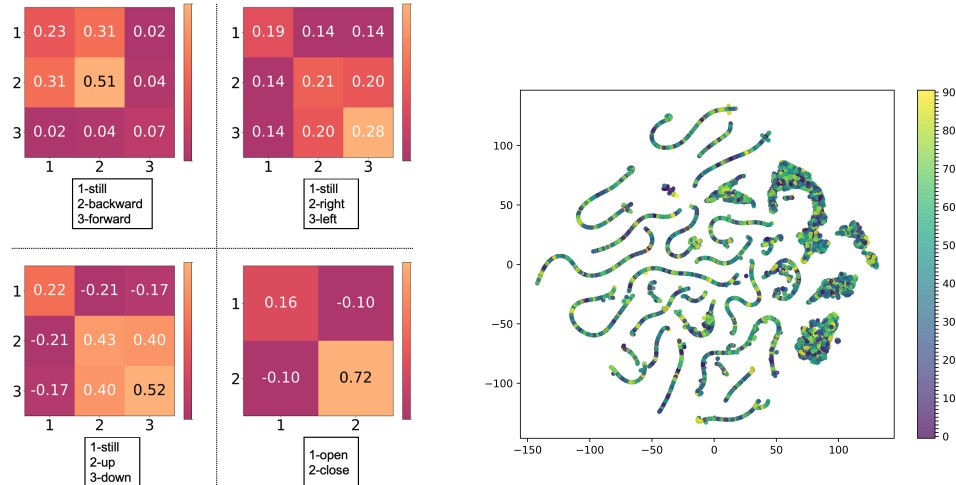

(a) Average cosine similarity between pairs of movement categories (defined by human semantics) computed using manipulation concept latents learned by our method (Sec.3).

(b) **t-SNE Clustering of Manipulation Concept Latents correpsonding to tasks.** We perform t-SNE clustering on the manipulation concepts at each time step. These concepts are generated by our method (Sec. 3). Each sample is colored according to its task, representing one of 90 possible tasks as indicated by the colorbar.

Figure 10

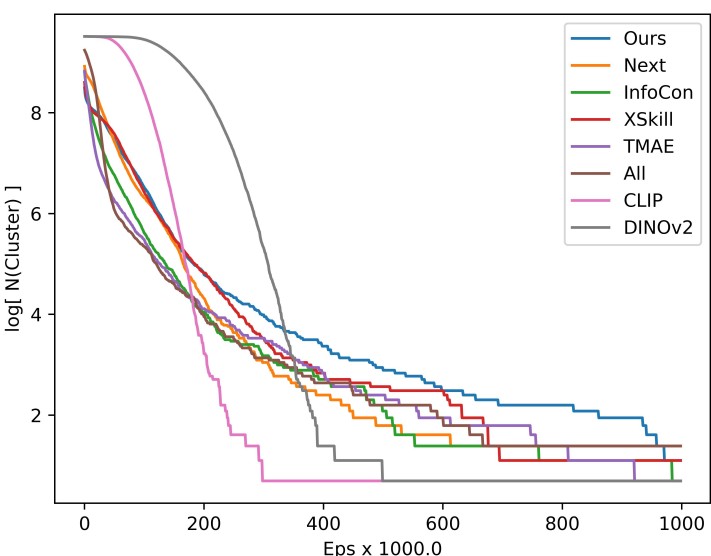

Figure 11: **DBSCAN Clustering Analysis of Manipulation Concept Latents' Diversity and Discrimination.** Clustering is performed on manipulation concept latents generated by our method and the baseline methods described in **Manipulation Concept Discovery Baselines** (Sec. 4.1), across 90 tasks from the LIBERO-90 dataset. The figure shows the (log) number of clusters obtained using DBSCAN for clustering density $\epsilon \in [0, 1]$, with no points classified as noise.

