# OpenReview forum: "$\textit{HiMaCon:}$ Discovering Hierarchical Manipulation Concepts from Unlabeled Multi-Modal Data"
_NeurIPS.cc/2025/Conference — NeurIPS 2025 poster_

### Official Review · Reviewer_7UNt · 2025-06-13

**Clarity:** 3
**Significance:** 3
**Originality:** 4
**Rating:** 5
**Confidence:** 4

**Summary:**

The paper proposes HiMaCon, a self-supervised framework for learning hierarchical manipulation concepts from unlabeled multi-modal robotic data. The method discovers latent representations that encapsulate both cross-modal correlations (e.g., between vision and proprioception) and multi-horizon temporal structure (short-term and long-term subgoals). These representations are then used to augment robotic manipulation policies, learnt from demonstrations using either a transformer based CVAE or a diffusion policy. Empirical evaluations on the LIBERO benchmark and real-world robot deployments demonstrate the method’s efficacy.

**Questions:**

- Why the choice of the spherical distance, as opposed to others, e.g. cosine similarity? Any reasons or intuition for this?
- If I read equation 4 / algorithm 1 correct, the same sub-process is defined as any pair-wise distance between to elements of the sub-process should be below the threshold? Why not just enforce a distance between the start and (current) end in Algorithm 1 while loop? Is that not sufficient?
- The z is used as an additional prediction signal for the policy to guide the learning. While this is clever and makes it easy to "add-on" to existing methods, it does mean that the policy network has to "re-learn" the structure captured by the Concept Encoder to correctly predict it. Did you try training a policy "head" on top of the Concept encoder directly, or an intermediate representation thereof?
- In the future work and appendix you refer to and compare to VLA models, and it seems that you come close to their performance. You mention potentially combining it with VLA models. Do you think the improved performance of a VLA is orthogonal to what HiMaCon provides to the ACT/DP models, i.e. that HiMaCon will offer additional improvements on a VLA, or would a VLA rather capture similar manipulation concepts in their intermediate layers and hence perform similar already? I'd love to hear the author's view on this.

**Ethical Concerns:**

["NO or VERY MINOR ethics concerns only"]

**Final Justification:**

Solid paper, and interesting points brought forward by the authors during the discussion, which can only improve the work.

**Limitations:**

in appendix due to space constraints, but yes

**Paper Formatting Concerns:**

I'm less fan of the in-text tables and figures. Maybe putting e.g. Table 3 and Figure 3 side by side could be an option. Minor comment however, and I fully understand this is due the strict 9p constraint.

**Quality:**

3

**Strengths And Weaknesses:**

*Strengths*
- The method is novel and well-motivated from physiological evidence. Whereas I have seen skill discovery using mutual information between (discrete) latent and observations before (e.g. https://arxiv.org/abs/2211.13350), exploiting cross-modal correlations is nice and elegant.
- I especially like the fact that it can be used as an "add-on" for existing methods, as evidenced by combinging it with both ACT and DP.
- Strong benchmark results on LIBERO and in addition nice real-world experiments and interpretability results.

*Weaknesses*
- The approach needs access to aligned modality (i.e. RGB and proprioceptive) as well as action information of demonstrations to train on, which might be costly to obtain for different use cases.

---

> ### Author Rebuttal · Authors · 2025-07-31
>
> Thank you for the thoughtful review. We appreciate your recognition of HiMaCon's novelty, particularly your observation that exploiting cross-modal correlations is "nice and elegant." We're pleased you found value in our approach's compatibility as an "add-on" for existing architectures like ACT and Diffusion Policy, which was a key design for practical applicability. Your appreciation of our comprehensive evaluation—including LIBERO benchmark results, real-world experiments, and interpretability analyses—is encouraging. We address your specific concerns below.
>
> ***
>
> # Re W1: Multi-Modal Data Collection Cost
>
> We acknowledge this is a valid practical consideration. However, we would like to highlight several points that address this concern:
>
> **Comparable Requirements to Standard Approaches**: Multi-modal data collection (vision + proprioception + actions) is standard in modern robotic platforms (e.g., ALOHA [1], UR, Franka) and datasets [2, 3]. Unlike methods requiring post-collection annotation (e.g., DecisionNCE baselines in Tab.1 require manual language labels), our framework is self-supervised, eliminating ongoing labeling costs.
>
> **Effectiveness with Reduced Sensor Requirements**: Our ablation studies (Tab.2) and no concept baseline (Plain in Tab.1) demonstrate that while additional modalities yield better performance, concepts learned from even single modalities provide policy improvements.
>
> **Validation with Cost-Effective Data Collection**: To further address cost concerns, we conducted an additional experiment on policies enhanced by concepts derived from human manipulation demonstrations collected via simple video recording for the real-world cleaning-cup task (Sec.4.4). This more accessible approach still enhances policy performance across scenarios:
>
> | | Place | Color | Obj. | Obst. | Barr. | Multi |
> |:-:|:-:|:-:|:-:|:-:|:-:|:-:|
> | w/o concept | 53.3% | 46.7% | 40.0% | 20.0% | 0.0% | 0.0% |
> | w/ concept (human) | 66.7% | 60.0% | 46.7% | 33.3% | 13.3% | 13.3% |
>
> These results demonstrate practical deployment potential under cost limitations while maintaining performance improvements.
>
> # Re Q1: Spherical Distance
>
> Thank you for this important question about our choice of spherical distance.
>
> **Empirical Validation**: We conducted ablation experiments comparing spherical distance with cosine similarity (cosine distance $\frac{1-\cos\theta}{2}$) as the coherence metric in Eq.4. The results demonstrate clear performance advantages for spherical distance:
>
> | Method | Cosine Similarity | Spherical Distance (Ours) |
> |:-:|:-:|:-:|
> | DP | 82.0±0.4 | 89.6±0.6 |
> | ACT | 67.8±0.5 | 74.8±0.8 |
>
> **Justification**: This empirical advantage aligns with the theoretical considerations that motivated our choice. The arccos transformation in spherical distance provides non-linear scaling that emphasizes differences between vectors with high cosine similarity, amplifying subtle distinctions in concepts that cosine similarity might miss.
>
> # Re Q2: Sub-process Segmentation Criterion
>
> Thank you for this insightful algorithmic question about our sub-process derivation strategy.
>
> ## Empirical Validation
> We tested the proposed start-to-end strategy on LIBERO-90 and found consistent performance degradation compared to our pairwise approach:
>
> | Method | Our Approach | Start-to-End Strategy |
> | :-: | :-: | :-: |
> | ACT | 74.8±0.8 | 68.4±0.8 |
> | DP  | 89.6±0.6 | 79.8±0.5 |
>
> ## Core Issue: Asymmetric Coherence
> The start-to-end approach enforces a "star pattern" where all concepts must be similar to the initial concept, but concepts later in the sequence may be quite different from each other. This can inappropriately group semantically distinct manipulation phases that happen to share similarity with an initial state.
>
> **Example**: Consider this manipulation sequence:
> ```
> [reach forward] → [reach upward] → [grasp cup] → [reach downward] → [place plate]
> ```
>
> If the coherence threshold allows "reach forward" to be similar to both "reach upward" and "reach downward" (due to shared movement characteristics), but "reach upward" and "reach downward" are dissimilar to each other, then:
>
> - **Start-to-end method**: Groups the entire sequence into one sub-process because each step is similar enough to the initial "reach forward"
> - **Our pairwise method**: Splits into `[reach forward → reach upward → grasp cup]` and `[reach downward → place plate]` because it detects the dissimilarity between upward and downward reaching
>
> **Why This Matters**: This asymmetric grouping can merge distinct manipulation sub-goals (e.g., "acquire object from top" vs. "place object at bottom"), thereby missing the discovery of meaningful hierarchical manipulation concepts.
>
> We acknowledge this merits further investigation and plan to explore additional sub-process derivation methodologies in future work (Sec.D).
>
> # Re Q3: Alternative Concept Integration Strategies
> Thank you for this insightful question about our architectural design choices.
>
> ## Empirical Comparison
> We compared using concept encoder features as direct input conditioning versus our joint prediction approach on LIBERO-90 tasks:
>
> | Method | Direct Conditioning | Joint Prediction (Ours) |
> |:-:|:-:|:-:|
> | ACT | 71.1±0.4 | 74.8±0.8 |
> | DP | 79.3±0.9 | 89.6±0.6 |
>
> Our joint prediction approach consistently outperforms direct conditioning across both policy architectures.
>
> ## Key Insight: Temporal Alignment
> The performance difference stems from a fundamental temporal alignment issue:
> - **Direct conditioning**: Uses concepts extracted from current/past observations as additional input features for predicting future actions, creating temporal misalignment.
> - **Joint prediction**: Learns to predict future concepts alongside future actions, creating temporal coherence between concept understanding and action planning.
>
> **Concrete Example**: In a cup placement task, after grasping an object, current concepts encode "grasping dynamics." However, for planning the next placement action, the policy needs to understand "placement dynamics." Our joint approach learns to predict placement-related concepts that directly inform upcoming actions, while direct conditioning provides historical grasping information that may be less relevant for placement planning.
>
> Nevertheless, direct conditioning on current/past manipulation concepts could prove valuable for tasks requiring explicit long-term memory [4], and we plan to explore hybrid approaches in future work.
>
> # Re Q4: HiMaCon and VLAs
>
> Thank you for this thoughtful question about the relationship between HiMaCon and VLA models. Our manipulation concepts offer **complementary benefits** to VLAs, particularly in training data efficiency.
>
> **Empirical Evidence**: We tested this hypothesis by integrating HiMaCon with OpenVLA-7B using the OpenVLA-OFT protocol [5] and the method in Sec.3.4. We fine-tuned on **half** of the LIBERO-10 demonstration data (for fair comparison, HiMaCon discovered concepts on the same LIBERO-10 data):
>
> | Method | w/o Concept | w/ Concept |
> |:-:|:-:|:-:|
> | Success Rate (%) | 81.0 | 90.2 |
>
> The improved performance under data-constrained conditions demonstrates that manipulation concepts provide valuable structural inductive biases for data-efficient policy learning.
>
> **Core Hypothesis**: HiMaCon captures fine-grained manipulation dynamics at multiple abstraction levels. Our concepts provide explicit intermediate representations that bridge high-level instructions and low-level actions. This reduces VLAs' learning burden by providing structured manipulation knowledge rather than requiring them to learn complex patterns from scratch.
>
> **Addressing Orthogonality**: With sufficiently large-scale training data, VLAs might internally develop similar manipulation representations simply using action prediction loss. However, robotic data collection faces fundamental scalability challenges. Our approach provides manipulation concepts, which may help VLAs derive similar representations with less data.
>
> **Additional Practical Advantages**: Our method can also extract concepts from non-robotic demonstration data (e.g., human videos) as demonstrated in **Re W1**, enabling diverse data sources to enhance VLA training.
>
> **Future Investigation**: We plan to analyze VLAs at different training scales to understand when explicit concept integration maintains advantages. We will also explore integration strategies for VLAs and concepts discovered by HiMaCon. While theoretical orthogonality remains an open question, we see clear practical value in manipulation concepts for data-efficient robotic learning.
>
> # Re Paper Formatting Concerns
>
> Thank you for this helpful formatting suggestion. We will implement the recommended side-by-side layout for Tab.3, Fig.3, and other similar cases.
>
> ***
>
> We sincerely appreciate the reviewer's feedback and hope our clarifications enhance understanding of our work's significance and presentation. Please don't hesitate to share any further thoughts or concerns.
>
> [1] Fu, Z., Zhao, T. Z., & Finn, C. (2024). Mobile aloha: Learning bimanual mobile manipulation with low-cost whole-body teleoperation. arXiv preprint arXiv:2401.02117.
>
> [2] Walke, H. R., Black, K., Zhao, T. Z., Vuong, Q., Zheng, C., Hansen-Estruch, P., ... & Levine, S. (2023, December). Bridgedata v2: A dataset for robot learning at scale. In Conference on Robot Learning (pp. 1723-1736). PMLR.
>
> [3] Khazatsky, A., Pertsch, K., Nair, S., Balakrishna, A., Dasari, S., Karamcheti, S., ... & Finn, C. (2024). Droid: A large-scale in-the-wild robot manipulation dataset. arXiv preprint arXiv:2403.12945.
>
> [4] Fang, H., Grotz, M., Pumacay, W., Wang, Y. R., Fox, D., Krishna, R., & Duan, J. (2025). sam2act: Integrating visual foundation model with a memory architecture for robotic manipulation. arXiv preprint arXiv:2501.18564.
>
> [5] Kim, M. J., Finn, C., & Liang, P. (2025). Fine-tuning vision-language-action models: Optimizing speed and success. arXiv preprint arXiv:2502.19645.

---

> > ### Comment · Reviewer_7UNt · 2025-08-01
> > **Thank you for the responses**
> >
> > I would like to thank the authors for addressing my questions, not only by answering them intuitively, but also backing them with further experimental evidence. This strengthens my belief that this work is strong enough to be accepted.

---

> > > ### Author Response · Authors · 2025-08-02
> > >
> > > Thank you for your feedback and evaluation. Your questions helped us strengthen both our experimental validation and presentation clarity. We appreciate your positive assessment of our work.

---

### Official Review · Reviewer_eio8 · 2025-07-03

**Clarity:** 3
**Significance:** 3
**Originality:** 3
**Rating:** 5
**Confidence:** 4

**Summary:**

This paper proposes a self-supervised framework to learn the hierarchical manipulation concepts. It operates through two mechanisms: 1) Cross-modal correlation learning to capture the invariant pattern across different sensory modalities, and 2) Multi-horizon sub-goal organization structures concepts hierarchically across temporal scales. The manipulation concepts can help imitation learning policy as an additional prediction goal. Experiments on simulation and real environment show the performance gain of the proposed method compared to multiple baselines.

**Questions:**

Please consider replying to the Weakness.

**Ethical Concerns:**

["NO or VERY MINOR ethics concerns only"]

**Final Justification:**

The authors solve my concerns in the rebuttal period.  Overall, I think this paper is well supported by extensive experiments and has great reproducibility, so I will keep my original rating and tend to accept it.

**Limitations:**

The authors have discussed the limitations.

**Paper Formatting Concerns:**

I have no formatting concerns.

**Quality:**

3

**Strengths And Weaknesses:**

Strength:

1. The proposed algorithm is well-motivated. It aligns with human intuition and intuitively can help to improve the generalization ability of manipulation policies. The method is also well designed. The two mechanisms have their own clear goals separately, which are also complementary.

2. The authors conduct extensive experiments in both simulation and real world to evaluate the proposed algorithm. The results show notable performance gain. Especially, I like the detailed analysis in Sec. 4.3 and 4.4 beyond pure quantitative numbers. These analysis greatly justify the algorithm design and help readers to understand the roles of each module.

3. The authors provide many details including pseudo code for their algorithm implementation, which would benefit the potential reproduction. The visualizations and figures are also clear.


Weaknesses:

1. I think this idea of manipulation concepts shares great similarity with skill learning since they both extract some high-level and generalizable information. There are already many previous works for skill learning like [r1]. I believe it would be helpful to discuss with these related works.

2. It is strange for me that the pretrained encoder is not re-used for the policy. Instead, the authors consider the manipulation concepts as extra regularization prediction goal. I hope the authors can give more explanation and comparison regarding this.

3. For cross-modal correlation, it is more intuitive to directly apply corss-modal contrastive learning like CLIP rather than this masked prediction to achieve the goal. Maybe the authors can give some disucssion about it.



[r1]: SkillDiffuser: Interpretable Hierarchical Planning via Skill Abstractions in Diffusion-Based Task Execution

---

> ### Author Rebuttal · Authors · 2025-07-31
>
> Thank you for the thoughtful and comprehensive review. We greatly appreciate your recognition of our work's motivation and the intuitive alignment with human manipulation concepts. Your acknowledgment that our two-mechanism approach (cross-modal correlation learning and multi-horizon sub-goal organization) is well-designed with clear, complementary goals is particularly encouraging, as this dual structure represents a core contribution of our framework. We are also grateful for your positive feedback on our extensive experimental validation across both simulation and real-world environments, and especially your appreciation of the detailed analyses in Sec.4.3 and Sec.4.4. Finally, thank you for noting our efforts to ensure reproducibility through detailed implementation descriptions and pseudocode. We address your concerns and questions below:
>
> ***
>
> # Re W1: Related Work of Skill Learning
> Thank you for identifying this important connection to the skill learning literature. We acknowledge that manipulation concepts and skill learning both aim to extract generalizable, high-level information from demonstrations, and we will expand our related work section to provide a more comprehensive discussion of skill learning approaches, including SkillDiffuser [r1] and other relevant works [r2, r3].
>
> Here we outline several distinctions between our approach and skill learning methods:
>
> **Conceptual Level**
>
> While skill learning focuses on discovering **executable action primitives**, our manipulation concepts capture more abstract patterns that inform policy learning:
>
> - **Skills** are executable action sequences that can be directly invoked (e.g., "grasp object", "move to location").
> - **Manipulation Concepts** capture manipulation-relevant patterns across multiple abstraction levels—from fine-grained motor actions to high-level subgoals that may combine multiple skills to achieve completion.
>
> **Technical Approach**
>
> - **Representation Space**: Unlike SkillDiffuser [r1] which uses discrete skill representations that may limit discoverable patterns, our continuous concept latents (lines 116-118) provide flexible representational capacity without finite codebook constraints.
> - **Multi-Modal Integration**: Our framework explicitly models cross-modal correlations through mask-and-predict strategies (Eq.3), enabling robust generalization across sensory variations—a focus less emphasized in typical skill learning approaches.
> - **Integration Mechanism**: Rather than requiring explicit skill selection as in [r1] or specialized hierarchical planning frameworks, our concepts seamlessly integrate with existing policy architectures through joint prediction (Eq.9), simply adding a concept prediction head ($\pi_z$) for regularization.
>
>
> # Re W2: Concept Encoder Usage
> Thank you for this insightful question about our architectural design choices. We tested both approaches during development and can provide empirical evidence supporting our design decision.
>
> ## Empirical Comparison
>
> We compared two conditioning strategies on LIBERO-90 tasks:
> - **Direct Conditioning**: Using the pretrained concept encoder $\mathcal{E}$ (Eq.1) to extract concepts from observations, then feeding these as additional input features to the policy
> - **Joint Prediction (Ours)**: Training the policy to predict future manipulation concepts alongside actions through joint optimization (Eq.9)
>
> | Method | Direct Conditioning | Joint Prediction (Ours) |
> |:-:|:-:|:-:|
> | ACT | 71.1±0.4 | 74.8±0.8 |
> | DP | 79.3±0.9 | 89.6±0.6 |
>
> Our joint prediction approach consistently outperforms direct conditioning.
>
> ## Key Insight: Temporal Alignment
>
> We believe the performance difference stems from a fundamental temporal alignment issue:
>
> - **Direct conditioning**: Uses manipulation concepts extracted from current/past observations as additional input features for predicting future actions, which creates a temporal misalignment
> - **Joint prediction**: Trains the policy to predict future manipulation concepts alongside future actions, creating temporal coherence between concept understanding and action planning
>
> **Concrete Example**: In a cup placement task, after grasping an object, the current manipulation concepts encode "grasping dynamics." However, for planning the next placement action, the policy needs to understand "placement dynamics." Our joint approach learns to predict placement-related concepts that directly inform upcoming actions, while direct conditioning provides historical grasping information that may be less relevant for placement planning.
>
> Nevertheless, we acknowledge that direct conditioning on current/past manipulation concepts could prove valuable for tasks requiring explicit long-term memory [r4], and we plan to explore hybrid approaches in future work.
>
> # Re W3: Masked Prediction instead of Contrastive Learning
>
> Thank you for this question about our choice of masked prediction over contrastive learning approaches like CLIP. We carefully considered both options and selected masked prediction primarily to avoid the inherent challenges and potential biases associated with negative sampling strategies in contrastive learning.
>
> Specifically, manipulation involves recurring conceptual patterns (approach, contact, grasp, transport, release) that can appear across different demonstrations or multiple times within the same demonstration. This creates significant challenges for principled negative sampling without manual annotation or external supervision.
>
> In contrast, our masked prediction approach elegantly sidesteps these challenges by providing self-supervised learning without requiring explicit similarity judgments. The model learns cross-modal correlations by reconstructing masked modalities using available context, naturally capturing the relationships essential for manipulation understanding. This approach eliminates the risk of conflicting supervision signals that could arise from improperly chosen negative samples. Therefore, we selected the mask-and-predict framework.
>
> ***
> Once again, we thank the reviewer for the detailed examination and hope our responses strengthen both the clarity and impact of our contributions. We invite any additional comments or questions you might have.
>
> [r1] Liang, Z., Mu, Y., Ma, H., Tomizuka, M., Ding, M., & Luo, P. (2024). Skilldiffuser: Interpretable hierarchical planning via skill abstractions in diffusion-based task execution. In Proceedings of the IEEE/CVF Conference on Computer Vision and Pattern Recognition (pp. 16467-16476).
>
> [r2] Rho, S., Smith, L., Li, T., Levine, S., Peng, X. B., & Ha, S. Language Guided Skill Discovery. In The Thirteenth International Conference on Learning Representations.
>
> [r3] Chen, L., Bahl, S., & Pathak, D. (2023, December). Playfusion: Skill acquisition via diffusion from language-annotated play. In Conference on Robot Learning (pp. 2012-2029). PMLR.
>
> [r4] Fang, H., Grotz, M., Pumacay, W., Wang, Y. R., Fox, D., Krishna, R., & Duan, J. (2025). Sam2act: Integrating visual foundation model with a memory architecture for robotic manipulation. arXiv preprint arXiv:2501.18564.

---

> > ### Comment · Reviewer_eio8 · 2025-08-05
> >
> > I appreciate the authors' efforts in the rebuttal period. This detailed response can solve most of my concerns. I will keep my original rating and I think this paper can be accepted.

---

> > > ### Author Response · Authors · 2025-08-05
> > >
> > > Thank you for your evaluation and for maintaining your support for our work. We're pleased that our detailed rebuttal successfully addressed your concerns. Your constructive feedback is invaluable in strengthening our paper, and we greatly appreciate your recommendation for acceptance.

---

### Official Review · Reviewer_WGug · 2025-07-03

**Clarity:** 3
**Significance:** 4
**Originality:** 3
**Rating:** 5
**Confidence:** 4

**Summary:**

This paper proposes a self-supervised pipeline to discover 'concepts' from a dataset of demonstrations by cross-modal alignment and temporal clustering, and use these concepts to help imitation learning by adding an auxiliary goal of concept prediction.

**Questions:**

* Please refer to weaknesses.
* The reviewer wonders how robust the pipeline is against visual disturbances, like background, lighting conditions, etc, and cross-embodiment generalization. The dataset used for concept discovery is quite similar to the test setting (all from Libero). Ideally, I think self-supervised training with any robotic dataset (even human manipulation videos from the internet) would help. Can the authors provide some intuition and, what would be even better, some preliminary empirical results?

If the authors can address the questions above, the reviewer will raise the score (possibly by a large margin).

**Ethical Concerns:**

["NO or VERY MINOR ethics concerns only"]

**Final Justification:**

The authors addressed all my concerns during the rebuttal period: (1) some closely related works and baselines were missing; (2) an ablation to support their claim; (3) generalization and robustness of their method. Therefore, given the novelty and technical solidness of this work, I have raised my score from 3 to 5.

**Limitations:**

yes.

**Paper Formatting Concerns:**

N/A.

**Quality:**

4

**Strengths And Weaknesses:**

Strengths:
* The idea of cross-modal alignment is natural and well motivated, and the experiment results are promising.

Weaknesses:
* There are some related works missing. For example, [1] has a very similar problem setting and method design; [2, 3] also learn concepts (object contact and geometrical relationships) from demonstrations. The author should consider comparing with [1] and adding all of the above to the literature review.
* In  Section 3.3, the authors say they are learning 'sub-goals' and claim that this is helpful for later manipulation learning. However, it is not clear to me whether this help only comes from planning in the latent space, or planning in **the very** latent space which incorporates cross-modal alignment. The author should consider a finetuned version of DINOv2 or CLIP with their original loss function plus the multi-horizon sub-goal loss function proposed in Eq. 7. I believe this ablation will help readers understand the crucial role of the method proposed in Section 3.2.

[1] Zhou, Pei, et al. "AutoCGP: Closed-Loop Concept-Guided Policies from Unlabeled Demonstrations." The Thirteenth International Conference on Learning Representations.
[2] Mao, Jiayuan, et al. "Learning reusable manipulation strategies." Conference on Robot Learning. PMLR, 2023.
[3] Liu, Yuyao, et al. "One-Shot Manipulation Strategy Learning by Making Contact Analogies." ICRA, 2025.

---

> ### Author Rebuttal · Authors · 2025-07-31
>
> Thank you for the feedback and for recognizing the key contributions of our work. We appreciate that you found the cross-modal alignment approach natural and well-motivated, and we are encouraged by your assessment of our experimental results as promising. Your summary accurately captures our core methodology of discovering manipulation concepts through cross-modal alignment and temporal clustering, followed by their integration into imitation learning through auxiliary concept prediction. We are grateful for your constructive feedback on our work. Below we provide our responses to the weaknesses and questions you have raised.
>
> ***
>
> # Re W1: Related Works
> We sincerely thank the reviewer for highlighting these important related works. We agree these references significantly enrich the context of our research and will explicitly discuss and compare them in the revised version to clearly position our contributions.
>
> ## Regarding AutoCGP [1]:
>
> While AutoCGP shares our goal of learning manipulation concepts from demonstrations, our approach differs fundamentally in three key aspects:
>
> **Multi-modal Concept Discovery:** AutoCGP primarily relies on proprioceptive states for concept extraction, whereas our method leverages comprehensive multi-modal observations (vision + proprioception). Our ablation study (Tab.2) demonstrates that incorporating multiple modalities consistently improves performance.
>
> **Hierarchical Temporal Structure:** Our coherence-based clustering approach (Eq.4) enables simultaneous discovery of concepts at multiple temporal scales within a unified framework. This allows policies to reason about both immediate actions and long-term goals simultaneously, whereas AutoCGP focuses on a single level of temporal abstraction through trajectory segmentation.
>
> **Policy Integration Strategy:** We integrate concepts through regularization applied to hidden representations (Eq.9), maintaining compatibility with diverse architectures while introducing only minimal additional model complexity. In contrast, AutoCGP explicitly conditions policy heads on predicted concepts, requiring an additional concept generation model that substantially increases both model capacity and computational requirements.
>
> **Experimental Validation:** To substantiate these technical differences, we conducted comparative experiments on LIBERO-90 using three configurations:
>
> - **AutoCGP (original):** Uses AutoCGP's proprioceptive-based concept discovery method and integration strategy.
> - **AutoCGP (HiMaCon):** Our multi-modal concept discovery method integrated using AutoCGP's concept conditioning strategy.
> - **Ours (DP):** Our complete framework using Diffusion Policy as the base policy architecture. We compare with Diffusion Policy (DP) since AutoCGP employs DP as its policy head.
>
> | Method | AutoCGP (original) | AutoCGP (HiMaCon) | Ours (DP) |
> |:-:|:-:|:-:|:-:|
> | Success Rate (%) | 81.9±0.3 | 83.5±0.8 | **89.6±0.6** |
>
> The improvement when using our concepts with AutoCGP's integration method demonstrates the quality of our manipulation concepts, while our full method's superior performance validates both our concept discovery and integration strategies.
>
> ## Regarding \[2, 3]:
> We appreciate the reviewer highlighting these valuable methods, which indeed address critical aspects of manipulation concept learning—particularly object contact behaviors and geometric relationships. Unlike these approaches, which typically require explicit labeling or prior knowledge about contact states, our approach achieves fully self-supervised discovery of hierarchical manipulation concepts directly from unlabeled multi-modal data, without any semantic annotations. Nonetheless, the demonstrated capabilities in \[2] and \[3] regarding few-shot learning scenarios inspire promising directions for future exploration. Extending our method to efficiently handle few-shot learning contexts represents an exciting avenue we intend to pursue in subsequent work.
>
> In summary, we will expand the "Related Work" section of our manuscript to explicitly include detailed discussions and comparative analyses with these important related works \[1, 2, 3]. This addition will clearly articulate how our methodology advances beyond existing approaches in terms of novelty, generality, and empirical performance, further enhancing the clarity and comprehensiveness of our contribution.
>
> # Re W2: Ablation to Isolate Contributions
> We sincerely thank the reviewer for this insightful question that helps clarify the individual contributions of our framework's components. Following the reviewer's suggestion, we conducted additional experiments to isolate the contribution of our cross-modal correlation learning (Sec.3.2). We fine-tuned CLIP on LIBERO-90 data using the original text-image contrastive loss combined with our multi-horizon sub-goal loss (Eq.7), while excluding our cross-modal correlation methodology from Sec.3.2 (Eq.3). We also tested additional ablations to better understand component interactions:
>
> | Method | Cross-Modal (Eq.3) | Multi-Horizon (Eq.7) | ACT | DP |
> |:-:|:-:|:-:|:-:|:-:|
> | CLIP + Eq.7 (reviewer suggestion) | X | O | 66.4±0.5 | 81.6±0.6 |
> | Cross-modal only | O | X | 69.1±0.6 | 82.8±1.0 |
> | Multi-horizon only | X | O | 71.6±0.4 | 80.5±0.5 |
> | Ours (Full method) | O | O | **74.8±0.8** | **89.6±0.6** |
>
> These results reveal several key insights:
>
> - **Cross-modal correlation learning provides substantial benefits**: The improvements over CLIP+Eq.7 and the multi-horizon-only ablation demonstrate that our mask-and-predict strategy for capturing cross-modal patterns (Sec.3.2) contributes meaningfully beyond hierarchical temporal learning alone. The planning will benefit from a latent space which incorporates cross-modal alignment.
> - **Both components are necessary and complementary**: Neither cross-modal learning alone nor multi-horizon prediction alone achieves full performance. This indicates that robust cross-modal correlations and multi-horizon subgoals have complementary effects, with their combination yielding superior performance compared to either component in isolation.
>
> We thank the reviewer for this valuable suggestion and will include these comprehensive ablations in our revision.
>
> # Re Q1: About Weaknesses
> Please refer to **Re W1: Related Works** and **Re W2: Ablation to Isolate Contributions**.
>
> # Re Q2: Robustness and Cross-Embodiment Generalization
> We thank the reviewer for this insightful question about robustness and cross-domain generalization. We conducted additional experiments to directly address these concerns.
>
> ## Visual Robustness Analysis
>
> We evaluated robustness to visual disturbances using our Real-world Generalization protocol (Sec.4.4) with two additional test conditions: (1) **Background variation**: placing distracting objects and patterns behind the manipulation workspace, and (2) **Lighting variation**: testing under different illumination conditions (bright, dim, and flash lighting). Results demonstrate that policies enhanced with our manipulation concepts show significantly improved robustness:
>
> |   | Background | Lighting |
> |:-:|:-:|:-:|
> | w/o concept | 33.3% | 53.3% |
> | w/ concept | 53.3% | 66.7% |
>
> ## Cross-Embodiment Concept Transfer
>
> We tested whether concepts learned from human demonstrations can enhance robotic policies. We collected human manipulation videos of the cleaning-cup task using a camera. We trained our concept discovery pipeline on these human videos and applied the learned concepts to enhance robotic policies:
>
> || Place | Color | Obj. | Obst. | Barr. | Multi | Background | Lighting |
> |:-:|:-:|:-:|:-:|:-:|:-:|:-:|:-:|:-:|
> | w/o concept | 53.3% | 46.7% | 40.0% | 20.0% | 0.0% | 0.0% | 33.3% | 53.3% |
> | w/ concept (human) | 66.7% | 60.0% | 46.7% | 33.3% | 13.3% | 13.3% | 40.0% | 60.0% |
>
> The consistent improvements demonstrate that our self-supervised framework captures manipulation patterns that transcend embodiment differences. This supports the reviewer's intuition about training with diverse data sources.
>
> We will include these results and video demonstrations in our revision.
>
> ***
>
> We deeply appreciate the reviewer's constructive comments and hope our responses provide clarity on the points raised. Please feel free to share any further concerns or suggestions you may have.
>
> [1] Zhou, Pei, et al. "AutoCGP: Closed-Loop Concept-Guided Policies from Unlabeled Demonstrations." The Thirteenth International Conference on Learning Representations.
>
> [2] Mao, Jiayuan, et al. "Learning reusable manipulation strategies." Conference on Robot Learning. PMLR, 2023.
>
> [3] Liu, Yuyao, et al. "One-Shot Manipulation Strategy Learning by Making Contact Analogies." ICRA, 2025.

---

> > ### Comment · Reviewer_WGug · 2025-08-01
> >
> > Thank the authors for the response. My concerns are addressed clearly. I will take the response into consideration in the later discussion with AC and final justification.
> >
> > I still have an additional question for the authors. In AutoCGP Section 4.2, they provided a qualitative visualization of the interpretability and consistency of the discovered concepts. I wonder whether the method proposed by the authors has similar properties, which I believe is an interesting thing to investigate.

---

> > > ### Author Response · Authors · 2025-08-02
> > >
> > > We sincerely thank the reviewer for the positive feedback and for raising this insightful question about interpretability and consistency analysis similar to AutoCGP [1].
> > >
> > > The manipulation concepts discovered by our method demonstrate interpretability and consistency properties, with the additional advantage of multi-scale hierarchical organization. Here's our comprehensive analysis:
> > >
> > > ## **Semantic Alignment with Human Concepts (Sec.4.3 Fig.3 & Sec.C.2 Fig.8)**
> > >
> > > We evaluate whether our learned concepts align with human-interpretable manipulation primitives.
> > >
> > > **Key Principle:** Semantically meaningful concepts should exhibit structured similarity patterns that respect human conceptual boundaries.
> > >
> > > **Methodology:** We group manipulation concept latents by human-annotated sub-goals (e.g., "grasp bowl," "open drawer," "pull drawer") and compute inter-group similarities using Eq.10. If our concepts capture human-interpretable semantics, then latents from the same category should be more similar to each other than to different categories.
> > >
> > > **About Fig.3 and Fig.8:** Each heatmap represents similarity matrices of different human-annotated sub-goals based on the manipulation concept latents. For each heatmap, the i-th row and j-th column indicate the inter-group similarity between human-annotated sub-goal #i and #j based on the manipulation concept latents.
> > >
> > > **Key Findings:**
> > >
> > > - **Diagonal Dominance:** Heatmaps in Fig.3 and Fig.8 consistently show the highest values along the diagonal, indicating that **concept latents from the same category are more similar to each other than to different categories**. This demonstrates successful capture of human-interpretable manipulation primitives without explicit supervision.
> > > - **Cross-Task Consistency:** Since human labels are shared across multiple tasks (detailed in lines 858-861, Sec.C.2), this analysis demonstrates that our concepts **capture consistent behaviors across different demonstrations and environments**.
> > > - **Multi-Granularity Semantics:** Our concepts capture structure at different abstraction levels. Fig.8 shows this hierarchy: fine-grained concepts include detailed information like spatial relationships (e.g., "open top drawer" vs "open bottom drawer" in the top heatmap), while coarse-grained concepts capture general manipulation goals (e.g., "open drawer" regardless of location in the bottom heatmap).
> > >
> > > ## **Hierarchical Structure Interpretability (Sec.C.5 Fig.6 & `supplementary/#0-Supplementary.pdf` Fig.2-7)**
> > >
> > > Our method provides interpretable hierarchical decomposition through the coherence threshold $\epsilon$ (Eq.4), as shown in Fig.6 in Sec.C.5 and Fig.2-7 in `supplementary/#0-Supplementary.pdf`. This offers unique insights into manipulation tasks at multiple temporal scales:
> > >
> > > - **Multi-Scale Organization:** Varying $\epsilon$ naturally segments demonstrations into interpretable sub-processes:
> > >     - **Large $\epsilon$:** Long-horizon coarse-grained phases.
> > >     - **Small $\epsilon$:** Short-horizon fine-grained subprocesses.
> > >
> > > - **Human-Interpretable Segments:** The derived sub-processes correspond to naturally describable manipulation concepts, as shown in the captions of these figures. This demonstrates that our representations capture semantic task structure at multiple abstraction levels.
> > >
> > > - **Cross-Environment Consistency:** Fig.6 and Fig.7 provide evidence of **hierarchical consistency across varied conditions**. For example, Fig.6 shows three pick-and-place demonstrations in different environments with different objects and backgrounds. Despite these variations, applying Eq.4 produces similar hierarchical decompositions:
> > >     - **Row 1:** "grasp object → place object"
> > >     - **Row 2:** "grasp → transition → release"
> > >     - **Row 3:** "approach → grasp → transition → release"
> > >
> > > ## **Comparison to AutoCGP**
> > > Our interpretability analysis provides comparable qualitative insights to AutoCGP with a key advantage: hierarchical organization. Unlike AutoCGP's single-level concepts, our method captures multiple temporal scales, allowing analysis of manipulation tasks at different levels of abstraction from fine-grained actions to coarse task phases.
> > >
> > > ## Summary
> > > Our interpretability analysis demonstrates that learned concepts capture meaningful semantics and show consistent structure across tasks and environments, providing interpretability comparable to AutoCGP while offering additional hierarchical insights into manipulation task structure.
> > >
> > > We thank the reviewer again for the insightful and encouraging comments. Please feel free to reach out if you have any further questions or would like additional clarification.
> > >
> > > [1] Zhou, P., Liu, R., Luo, Q., Wang, F., Song, Y., & Yang, Y. AutoCGP: Closed-Loop Concept-Guided Policies from Unlabeled Demonstrations. ICLR 2024.

---

> > > > ### Comment · Reviewer_WGug · 2025-08-07
> > > >
> > > > Thank the authors for their effort during the rebuttal period and for addressing all of my concerns. As I promised in my original review, I have raised the score of this paper from 3 to 5.

---

> > > > > ### Author Response · Authors · 2025-08-07
> > > > >
> > > > > Thank you for taking the time to carefully review our rebuttal and for your constructive feedback throughout the process. We greatly appreciate your willingness to reconsider our work and are grateful for the score increase. Your insights have helped strengthen our paper.

---

### Official Review · Reviewer_6hJG · 2025-07-03

**Clarity:** 3
**Significance:** 4
**Originality:** 3
**Rating:** 5
**Confidence:** 4

**Summary:**

This paper introduces HiMaCon, a self-supervised framework designed to learn hierarchical concepts for robotic manipulation from unlabeled multi-modal data. The authors propose that learning abstract, hierarchically structured concepts can bridge the gap between low-level actions and high-level goals, enhancing policy robustness. The HiMaCon framework achieves this through two complementary mechanisms:

(1) a cross-modal correlation network that learns invariant relationships between different sensory inputs (e.g., vision and proprioception),

(2) and a multi-horizon predictor that organizes these concepts into a temporal hierarchy of short-term and long-term sub-goals.

The paper demonstrates through extensive experiments in both simulation and the real world that policies enhanced with these learned concepts significantly outperform standard imitation learning approaches, especially in challenging generalization tasks. The analysis also reveals that the discovered concepts are semantically meaningful and align with human-interpretable manipulation primitives, despite the absence of explicit labels during training.

**Questions:**

1. The sub-process derivation via the coherence threshold $\varepsilon$ is a cornerstone of your method. Could you provide more intuition on how the uniform sampling of $\varepsilon \sim \mathcal{U}(0,1)$ during training encourages the emergence of a meaningful and stable hierarchy? Did you experiment with alternative sampling strategies (e.g., discrete levels, a biased distribution) or analyze the sensitivity of the learned representations to this choice?
2. In your real-world experiments, you hypothesize that concept enhancement provides a "relational focus" that allows the policy to ignore superficial features. This is a powerful claim. Can you elaborate on the mechanism by which the cross-modal correlation objective (Eq. 3) specifically encourages the model to learn abstract relations (like "object inside container") over concrete visual features (like object color or shape)? Is it because relational properties have more consistent correlations across modalities than surface features do?
3. For real robot experiments, you only report success rates. Could you provide a more detailed analysis of failure modes? How does the concept-enhanced policy's behavior differ from the baseline in failed attempts?

**Ethical Concerns:**

["NO or VERY MINOR ethics concerns only"]

**Final Justification:**

The authors addressed my concerns well in the rebuttal period with more supportive experiments. I will keep my original rating and tend to accept it.

**Limitations:**

Yes.

**Paper Formatting Concerns:**

No major flaws.

**Quality:**

3

**Strengths And Weaknesses:**

=== Strengths ===

1. The combination of cross-modal correlation learning and multi-horizon sub-goal organization is interesting. The mask-and-predict strategy for capturing cross-modal patterns and the coherence-based clustering for temporal hierarchy are well-motivated.
2. The paper provides extensive experiments across multiple settings (LIBERO-90, LIBERO-LONG, LIBERO-GOAL) and includes both simulation and real-world validation. The proposed method consistently outperforms baselines across different policy architectures (ACT and Diffusion Policy). The performance gains are particularly notable in transfer and generalization scenarios.
3. The semantic alignment study showing that learned concepts correspond to human-interpretable sub-goals is valuable. The visualization of multi-horizon predictions provides good insights into what the model learns.

=== Weaknesses ===

1. The concept discovery phase involves training multiple transformer-based networks, which appears to be computationally intensive.
2. The derivation of multi-horizon sub-processes relies on a coherence threshold $\varepsilon$ applied to the cosine distance between latent concepts (Eq. 4). While sampling $\varepsilon$ uniformly during training is a reasonable strategy, the underlying mechanism feels somewhat heuristic. The paper could be strengthened by discussing the sensitivity to this mechanism or exploring alternative, more structured approaches to discovering the hierarchy.

---

> ### Author Rebuttal · Authors · 2025-07-31
>
> Thank you for your thoughtful and comprehensive review. We appreciate your recognition of the key technical contributions, particularly the novelty of combining cross-modal correlation learning with multi-horizon sub-goal organization. Your acknowledgment of our extensive experimental validation across multiple benchmarks and policy architectures, as well as the semantic interpretability analysis, is very encouraging. We're glad that you found the mask-and-predict strategy and coherence-based clustering well-motivated, and that you recognized the value of our semantic alignment study in demonstrating that HiMaCon learns meaningful manipulation primitives. Below we provide our responses to the weaknesses and questions you have raised.
>
> ***
>
> # Re W1: Computation
>
> We acknowledge the reviewer's concern about computational requirements and appreciate the opportunity to clarify this aspect of our method.
>
> **Reasonable Computational Requirements:** Our experiments demonstrate that the complete manipulation concept discovery pipeline operates under accessible computational constraints. As detailed in Sec.A.1, the concept discovery training can be executed on widely available hardware such as a single RTX 3090/4090 GPU (24GB memory), completing within 1.5 days. The architecture is deliberately modest, consisting of a 12-layer concept encoder $\mathcal{E}$ and two 4-layer networks for cross-modal correlation $\mathcal{C}$ (Eq.3) and multi-horizon prediction $\mathcal{F}$ (Eq.7), totaling approximately 48M parameters across all three networks with 256 hidden dimensions.
>
> **Strong Return on Investment:** This represents a one-time computational investment. For example, once trained on LIBERO-90, the learned concepts transfer directly to LIBERO-LONG and LIBERO-GOAL without retraining (Sec.4.2), while consistently delivering performance improvements across benchmarks (Tab.1).
>
> **Future Optimization:** We acknowledge the importance of computational efficiency for larger-scale deployment. We are actively exploring strategies including foundation model initialization with efficient fine-tuning techniques like LoRA, and alternating training strategies between cross-modal correlation learning (Eq.3) and multi-horizon sub-goal learning (Eq.7) to reduce peak memory requirements.
>
> # Re W2 and Q1: Sampling Strategies
>
> We thank the reviewer for their insightful questions about our hierarchical derivation mechanism and the sampling strategy of $\epsilon$ during training.
>
> **Uniform Sampling for Hierarchical Representation Learning**
>
> Our approach implements a form of multi-scale temporal abstraction learning. Different $\epsilon$ values correspond to different levels of temporal compression: low $\epsilon$ values identify fine-grained manipulation primitives, while high $\epsilon$ values discover coarse-grained subprocesses. By considering all possible $\epsilon$ across this spectrum, we ensure that learned concepts can encode subgoals that are maximally informative about future state transitions across all temporal scales (Eq.5).
>
> The uniform sampling of $\epsilon\sim\mathcal{U}(0,1)$ is motivated by computational considerations. While considering all possible $\epsilon$ values during each training iteration would be ideal, this approach is computationally prohibitive, leading us to employ uniform sampling as a practical approximation.
>
> **Why Uniform Distribution Rather Than Alternatives**
>
> Since different manipulation tasks naturally exhibit varying optimal temporal structures, uniform sampling provides equal representational capacity across all granularities. This prevents the model from biasing toward any particular temporal scale, which could lead to suboptimal hierarchy derivation for certain tasks.
>
> **Experimental Validation**
>
> We conducted experiments on LIBERO-90 to validate our design choice, testing multiple alternative strategies:
>
> | Sampling Strategy | ACT | DP | Description |
> |-|-|-|-|
> | **Uniform (Ours)** | **74.8±0.8** | **89.6±0.6** | $\epsilon\sim\mathcal{U}(0,1)$ |
> | Discrete | 67.6±0.5 | 81.1±0.8 | $\epsilon\sim${$0.1, 0.2, ..., 1.0$} |
> | Biased | 65.6±0.7 | 78.7±0.4 | $\epsilon\sim\mathcal{U}(\frac{1}{3},\frac{2}{3})$ |
>
> We will provide additional studies, experiments, and analysis of different selection strategies in the revised version, as we acknowledge that the sub-process derivation strategy is essential for the quality of learned concepts.
>
> # Re Q2: Mechanism for Relational Learning
>
> Thank you for this insightful question about the mechanism underlying our cross-modal correlation objective. We explain how Eq.3 encourages learning abstract relations over superficial features through a principled information-theoretic argument.
>
> **The Fundamental Challenge: What Information Enables Cross-Modal Prediction?**
>
> Consider a robot placing an object in a container, observed through both a gripper-mounted camera (top view) and a third-person camera (front view). Our cross-modal correlation objective requires the model to predict one view from the other, conditioned on the manipulation concept latent. This creates a fundamental question: what type of information must the concept latent encode to successfully bridge between these different visual perspectives?
>
> **Why Surface Features Cannot Solve Cross-Modal Prediction**
>
> The answer emerges from analyzing what information is already available versus what is missing. Both camera views already contain comprehensive appearance information—colors, textures, and object shapes are visible in both modalities. Therefore, if the concept latent redundantly encodes this already-available appearance information, it provides no additional predictive power for cross-view generation. Surface features like "object A is red" offer minimal benefit when predicting the front camera view from the top camera view, since color is already observable in the source view.
>
> **Why Relational Properties Are Essential**
>
> In contrast, relational properties like spatial arrangements and containment relationships ("object A is inside container B") provide the geometric understanding necessary to successfully map between different visual perspectives. These abstract relationships remain consistent across viewpoints but require sophisticated understanding to bridge the visual gap between perspectives. From an information-theoretic standpoint, relational properties exhibit significantly higher mutual information across modalities than surface features, making them the viable solution for cross-modal prediction.
>
> **The Representational Bottleneck Effect**
>
> This analysis reveals the core mechanism: our objective creates a "representational bottleneck" that acts as a natural filter. Since surface features cannot solve the cross-modal prediction task, the learning process is forced to discover and encode view-invariant relational structure. The prediction objective thus transforms the inherent cross-modal consistency of relational properties into learned representational structure that prioritizes relational understanding over superficial appearance.
>
> # Re Q3: Failure Modes
>
> Thank you for this important question. We provide a failure mode analysis based on examination of failed attempts across our real robot experiments, revealing how manipulation concepts change robot behavior patterns.
>
> - **Subtask Awareness and Persistence:** Baseline failures frequently exhibit "phantom grasping"—proceeding to move toward containers without successfully grasping objects, indicating poor subtask completion awareness. In contrast, concept-enhanced failures demonstrate persistent re-grasping attempts when initial grasp attempts fail (1-4 times), maintaining task coherence by recognizing incomplete subgoals. While these attempts ultimately fail due to time constraints or object displacement, the policies show systematic task progression awareness rather than premature abandonment.
>
> - **Goal-Directed vs. Wandering Behavior:** In the **Barriers** condition, failure patterns diverge markedly. Baseline failures exhibit "wandering" behavior after grasping—moving the gripper aimlessly around containers without structured placement attempts, suggesting complete loss of task direction. Concept-enhanced failures maintain goal-directed behavior, successfully navigating barriers but failing due to physical constraints such as container tipping from elbow contact. Crucially, these policies continue executing placement subgoals rather than abandoning task progression.
>
> **Implications**: These patterns demonstrate that concept prediction training (Eq.9) provides policies with enhanced understanding of task progress and subgoal states. Concept enhancement transforms failure modes from random task abandonment to systematic, goal-directed behavior that maintains task coherence even when ultimately unsuccessful.
>
> We will include these detailed behavioral comparisons in the revised version of the paper.
>
> ***
>
> We thank the reviewer again for these thoughtful questions and hope our clarifications reinforce the value and clarity of our contributions. Please feel free to let us know if you have more suggestions or questions.

---

> > ### Comment · Reviewer_6hJG · 2025-08-05
> >
> > I thank the authors for their significant effort in rebuttal and conducting extra experiments. After reading the rebuttal, my concerns are well addressed. I will keep my initial rating of acceptance.

---

> > > ### Author Response · Authors · 2025-08-05
> > >
> > > Thank you for your review and for recognizing our efforts in the rebuttal process. We're grateful that our additional experiments and responses effectively addressed your concerns. We appreciate your continued support and recommendation for acceptance.

---

### Note · Authors · 2025-08-12

We are deeply grateful to all reviewers for their thorough evaluation and constructive feedback, which has significantly strengthened our work. We sincerely thank Reviewer 6hJG for acknowledging the novelty of combining cross-modal correlation learning with multi-horizon sub-goal organization and for recognizing both our extensive experimental validation across multiple benchmarks and policy architectures and our semantic interpretability analysis. We appreciate Reviewer WGug for finding our cross-modal alignment approach well-motivated. We are grateful to Reviewer eio8 for recognizing our work's intuitive motivation and the complementary design of our dual-mechanism approach, and for appreciating our detailed analyses and reproducibility efforts. Finally, we thank Reviewer 7UNt for highlighting the elegance of exploiting cross-modal correlations and recognizing the practical value of our "add-on" compatibility with existing architectures.

To address reviewer concerns, we conducted additional experiments including computational analysis, ablation studies, robustness evaluation, and comparative analysis:

- **Reviewer 6hJG**: We justified computational requirements and validated our uniform sampling strategy against alternatives. We also provided extra discussion on cross-modal learning mechanisms and failure mode analysis.

- **Reviewer WGug**: We performed ablation studies isolating cross-modal contributions, added comparisons with missing works including AutoCGP, and demonstrated robustness across visual disturbances and cross-embodiment transfer. We provided interpretability evidence showing semantic alignment and hierarchical consistency comparable to AutoCGP while offering unique multi-scale insights.

- **Reviewer eio8**: We clarified distinctions from skill learning literature, provided empirical justification for joint prediction over direct conditioning, and explained our masked prediction choice over contrastive learning.

- **Reviewer 7UNt**: We addressed data collection costs, validated spherical distance over cosine similarity, justified our pairwise sub-process derivation, and demonstrated HiMaCon's complementary benefits with VLA models.

Once again, we sincerely thank all reviewers for their valuable feedback and thoughtful suggestions toward enhancing the paper.

---

### Decision · Program_Chairs · 2025-09-17

**Decision:**

Accept (poster)

**Comment:**

All reviewers acknowledge the novelty of the proposed method and its superior performance compared to the DP and ACT baselines. However, the paper does not compare against numerous existing works that also predict video frames or other forms of subgoals, nor against several methods that achieve significantly stronger performance on the LIBERO benchmark—such as Flower-VLA (https://openreview.net/pdf?id=ifo8oWSLSq).